# The evolutionary origin of naturally occurring intermolecular Diels-Alderases from *Morus alba*

Qi Ding [1,2,9], Nianxin Guo [2,3,4,9], Lei Gao [2] ✉, Michelle McKee[5], Dongshan Wu [2], Jun Yang [2,3], Junping Fan [2], Jing-Ke Weng [5,6,7] & Xiaoguang Lei [2,3,4,8] ✉

Biosynthetic enzymes evolutionarily gain novel functions, thereby expanding the structural diversity of natural products to the benefit of host organisms. Diels-Alderases (DAs), functionally unique enzymes catalysing [4 + 2] cycloaddition reactions, have received considerable research interest. However, their evolutionary mechanisms remain obscure. Here, we investigate the evolutionary origins of the intermolecular DAs in the biosynthesis of Moraceae plant-derived Diels-Alder-type secondary metabolites. Our findings suggest that these DAs have evolved from an ancestor functioning as a flavin adenine dinucleotide (FAD)-dependent oxidocyclase (OC), which catalyses the oxidative cyclisation reactions of isoprenoid-substituted phenolic compounds. Through crystal structure determination, computational calculations, and site-directed mutagenesis experiments, we identified several critical substitutions, including S348L, A357L, D389E and H418R that alter the substrate-binding mode and enable the OCs to gain intermolecular DA activity during evolution. This work provides mechanistic insights into the evolutionary rationale of DAs and paves the way for mining and engineering new DAs from other protein families.

Natural products contain diverse biological and pharmaceutical activities due to their structural diversity[1]. The evolution and functional diversification of biosynthetic enzymes facilitate the generation of structural complexity and diversity of natural products[2]. The abundance of phylogenetically related but functionally diverse natural product biosynthetic enzymes provides opportunities for understanding the molecular mechanisms and evolutionary trajectories of newly emerged enzymes[3–5]. Knowledge learned from this effort will ultimately guide us in engineering new enzymes with desirable functions through directed evolution or rational design[6].

More than 150 different Diels–Alder (D-A)-type cycloadducts have been reported in natural products, including polyketides, isoprenoids, phenylpropanoids, and alkaloids[7]. Biosynthetically, these natural products are formed through Diels–Alder reactions, in which a [4 + 2] cycloaddition occurs between a conjugated diene and a dienophile. Several enzymes catalyzing Diels–Alder (D-A) reactions have been identified to date, such as the multifunctional DAs EupfF[8], SpnF[9,10], and LepI[11], and the monofunctional DAs PyrE3[12] and SdnG[13]. The previously discovered DAs belong to different protein families, such as FAD-dependent oxidases, SAM-dependent methyltransferases, lipocalins,

[1]School of Life Science, Tsinghua University, Beijing 100084, China. [2]Beijing National Laboratory for Molecular Sciences, Key Laboratory of Bioorganic Chemistry and Molecular Engineering of Ministry of Education, College of Chemistry and Molecular Engineering, Peking University, Beijing 100871, China. [3]Peking-Tsinghua Center for Life Sciences, Peking University, Beijing 100871, China. [4]Academy for Advanced Interdisciplinary Studies, Peking University, Beijing 100871, China. [5]Whitehead Institute for Biomedical Research, Cambridge, MA 02142, USA. [6]Institute for Plant-Human Interface, Northeastern University, Boston, MA 02120, USA. [7]Department of Chemistry and Chemical Biology and Department of Bioengineering, Northeastern University, Boston, MA 02120, USA. [8]Institute for Cancer Research, Shenzhen Bay Laboratory, Shenzhen 518107, China. [9]These authors contributed equally: Qi Ding, Nianxin Guo. ✉e-mail: gaolei0408@pku.edu.cn; xglei@pku.edu.cn

malate synthases, and polyketide synthases[14]. This diversity suggests that these DAs have evolved independently from different ancestral folds to gain specific DA activities. However, the evolutionary origins and mechanisms of natural DAs remain understudied.

Mulberry D–A-type adducts are a class of isoprenoid-substituted phenolic compounds unique to Moraceae plants[15]. Many mulberry D–A-type adducts exhibit promising medicinal properties, including anticancer and antiviral activities[16,17]. Our previous studies revealed that two berberine bridge enzyme (BBE)-like enzymes function sequentially in the biosynthetic pathway of D–A type natural product chalcomoracin. Among them, MaMO catalyzes an oxidative dehydrogenation reaction to produce the diene, which is further transformed into the D–A product via an intermolecular Diels–Alder reaction, catalyzed by the other BBE-like enzyme MaDA[18]. Unlike most BBE-like enzymes, which typically catalyze various oxidation reactions using FAD as a cofacor[19], such as oxidative cyclization by EcBBE[20], tetrahydrocannabinolic acid synthase (THCAS)[21], and cannabidiol acid synthase (CBDAS)[22], and oxidative dehydrogenation by MaMO and AtBBE-like 13[23], MaDA stands out because it catalyzes a non-redox intermolecular Diels–Alder reaction. How MaDA lost the redox activity and gained the new D–A function remains to be elucidated.

Here, we aim to elucidate the evolutionary origin of MaDAs in Moraceae plants. Utilizing phylogenetic analysis, ancestral sequence reconstruction (ASR), in vitro protein expression and functional characterization, protein structural determination, and structure-based mutagenesis, we found that these DAs originated from gene duplication followed by the neofunctionalization of BBE-like enzymes that gained oxidocyclase (OC) activity. Upon the emergence of several key substitutions, including S348L, A357L, D389E, and H418R, OCs further gained intermolecular DA activity during evolution. This study illustrates how newly functionalized DAs evolve from enzymes with oxidation and cyclization activities to achieve metabolic diversification in the biosynthesis of natural products. Through this work, we have expanded our understanding of the evolutionary rationale of intermolecular DAs and gained new insights for rationally mining more DAs from nature or designing artificial ones with desirable activities[24–26].

## Results

### FAD-dependent DAs in Moraceae plants evolve from the BBE-like enzymes

To elucidate the phylogenetic relationships of MaDA and other BBE-like enzymes, we thoroughly searched for proteins in this enzyme family. 7576 BBE-like enzymes were identified by mining a non-redundant protein database of all species as well as the transcriptomes and genomes of Moraceae species. A comprehensive phylogenetic analysis was performed (Supplementary Fig. 1), and a simplified phylogenetic tree was obtained accordingly (Fig. 1). Based on the classification of ref. 19, the phylogenetic tree was subdivided into six clades (5a-f). MaDA, MaMO, and an uncharacterized gene (CL164) form a new subclade, which is a sister to another subclade within the clade of 5d consisting of CBDAS[22], cannabichromene acid synthase (CBCAS)[27], and THCAS[28] (Fig. 1). This new subclade contains 185 putative genes that are all from Moraceae plants (Supplementary Table 1), indicating the phylogenetical uniqueness of these BBE-like enzymes. Further sequence analysis revealed that these genes were phylogenetically grouped into three clades (clades I-III), and duplication events for clade II and clade III occurred more recently than those leading to clade I. The Bayesian inference-based phylogenetic trees are topologically consistent with those constructed by maximum likelihood (Supplementary Fig. 2).

To functionally characterize these enzymes from clade I-III, 13 genes from *Morus alba*, including *CL164* from clade I, *MaDAs 4-8* from clade II, and *MaDSs 1-7* from clade III (Fig. 2), were cloned. Their encoded proteins were heterogeneously expressed and purified. Enzymatic assays show that the uncharacterized CL164 in clade I

functions like a cannabinoid oxidocyclase[29], catalyzing the formation of moracin D (**2**) from moracin C (**1**) via an oxidative cyclization reaction (Fig. 2). Thus, enzymes in this clade are named as *Morus alba* FAD-dependent oxidocyclases (OCs) and CL164 was named as MaOC1 accordingly. Like two previously reported DAs (MaDA[18] and MaDA1[30]), five enzymes in clade II (MaDA4-8) were also found to catalyze the [4 + 2] cycloaddition reaction of diene **3** and morachalcone A (**4**) to yield chalcomoracin (**5**) (Fig. 2), suggesting that the remaining uncharacterized enzymes in clade II are intermolecular DAs in the biosynthesis of Diels–Alder-type adducts in *Artocarpus heterophyllus*[31], *Morus nigra*[32], *Morus macroura*[33], and *Morus notabilis*[34] (Supplementary Table 1). Eight mulberry enzymes in clade III, including MaMO, showed oxidative activity on moracin C (**1**) to form diene **3** and thus named FAD-dependent diene synthases (DSs) (Fig. 2). Notably, two DAs (MaDA1 and MaDA4) and two DSs (MaDS1 and MaDS6) exhibit functional promiscuity, also functioning like OCs to produce moracin D (**2**) (Fig. 2b, c). In short, we have biochemically identified three types of BBE-like enzymes in *Morus alba*, namely OCs, DAs, and DSs. We found that some DAs and DSs showed oxidative cyclization activity towards moracin D (**2**).

The topological properties of the phylogenetic tree and the functional promiscuity of DAs imply that the oxidative cyclization activity might be the ancestral activity of DAs (Fig. 2). To test this hypothesis, we reconstructed the putative ancestral Diels-Alderases (ancDA) and the last common ancestor of DAs and DSs (ancDADS) by ASR (Fig. 2b). We then biochemically characterized their catalytic functions. Enzymatic assays showed that ancDA is functionally similar to MaDA1 and MaDA4. It is capable of catalyzing both the Diels–Alder reaction and the oxidative cyclization reaction of moracin C (**1**). AncDADS showed neither DA activity nor diene synthase activity but could catalyze the oxidative cyclization of moracin C (**1**) (Supplementary Fig. 3a, b). Interestingly, ancDADS could also oxidatively cyclize morachalcone A (**4**) to produce compound **6**, which was not observed in ancDA and extant DAs (Supplementary Fig. 3c), indicating that this oxidative cyclization function was completely lost during the evolution of moraceous DAs. In addition, the optimal reaction temperature of ancDADS was biochemically characterized as 70 °C, which was 20 °C higher than that of the extant enzyme MaDA[18] (Supplementary Fig. 3d, e). These results are consistent with previous reports showing that resurrected ancestral enzymes usually exhibit promiscuous activities and relatively high thermostability[35–38] and support the hypothesis that FAD-dependent DAs in Moraceae plants evolved from FAD-dependent OCs.

According to the enzyme evolution model proposed by Tawfik and colleagues[39], a novel enzymatic activity often emerges as a minor activity in the ancestral enzyme. While such activity can gradually be enhanced through natural selection, the original activity may be lost. In this study, some DAs and DSs retain their original oxidative cyclization activity to varying degrees (Fig. 2c). Thus, we propose that after gene duplication and subsequent mutations, ancestor OCs might undergo three evolutionary paths to generate functionally diverse DAs, DSs, and OCs, respectively (Fig. 2d): some of the enzymes retained their original oxidative cyclization activity; some of the enzymes evolved into DSs, laying the foundation for diene synthesis; some enzymes gradually lost their oxidative activities and became the standalone intermolecular DAs.

### Four activity-switching residues enable ancDADS to acquire DA activity

To gain insight into the structure-function relationships between OCs and DAs, we determined the crystal structure of a multifunctional Diels-Alderase MaDA1 (Fig. 3a). We also employed the AlphaFold method to model the structures of ancDA and ancDADS (Supplementary Fig. 4). Comparative analysis of the protein structures of MaDA1, ancDA, and ancDADS revealed a similar overall fold to BBE-like

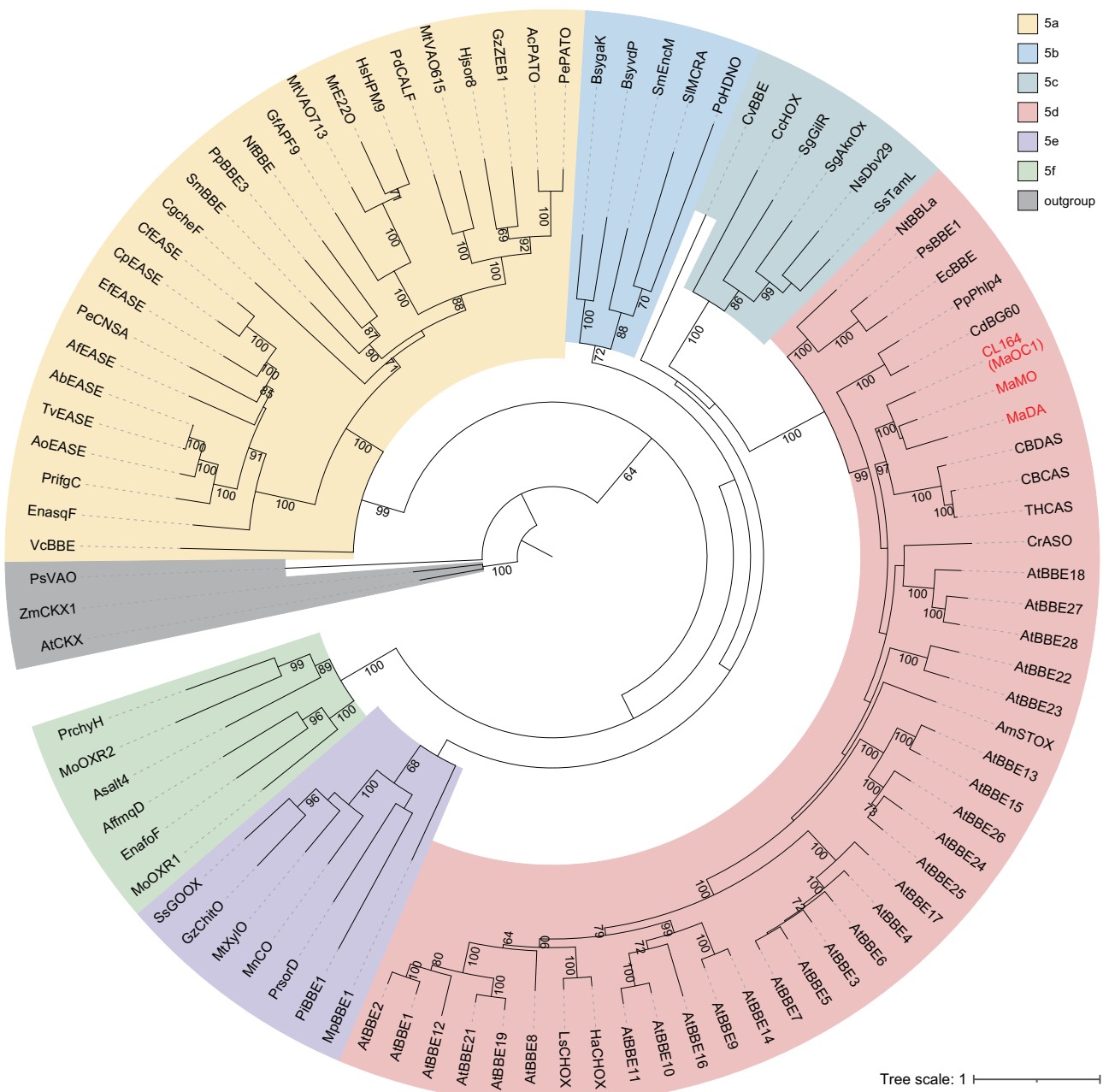

**Fig. 1 | The simplified phylogenetic tree of the BBE-like enzyme family.** The maximum likelihood method was used to construct the phylogenetic tree with 1000 bootstrap supports. Three FAD-linked oxidases, AtCKX (Q9FUJ1), ZmCKX1 (Q9T0N8), and PsVAO (P56216), were used as outgroups. Each clade corresponds to one color. The percent bootstrap values are shown for each clade with values greater than 60%.

enzymes, as previously reported[19]. Specifically, they possess six β-sheets (β12-14 and β16-18), one α-helix (α11), one loop (loop358-368 in MaDA1), and a cofactor FAD, and share a similar substrate-binding pocket among these enzymes (Fig. 3b). Notably, the structural analysis also suggested the volumes of the substrate-binding pockets in ancDADS, ancDA, MaDA, and MaDA1 are similar, indicating that no significant substrate-binding pocket expansion occurred during the evolution of the standalone DAs (Supplementary Fig. 5).

To identify residues important for D-A reaction activity, we docked the reported *endo*-transiton state (TS) of this D-A reaction[30] and conducted MD simulation experiments on ancDA (Fig. 3c) as well as MaDA1 (Supplementary Fig. 6a). Drawing from our previous research, we observed that in MaDA, R443 contributes to lowering the dienophile's LUMO energy by 0.8 eV through hydrogen bonding interactions, facilitating the Diels−Alder reaction. Additionally, E414

stabilizes these interactions by forming a hydrogen bond with R443[30]. As similar interactions were also noted in ancDA, we hypothesize that R418 and E389 in ancDA play analogous roles in catalyzing the Diels−Alder reaction, similar to the functions of R443 and E414 in MaDA (Fig. 3c). Similar results were observed in the computational studies of MaDA1 (Supplementary Fig. 6a). Consistent with the computational findings, these two residues were found to be evolutionarily conserved in DAs, while all OCs, including ancDADS, contain a D and H at the corresponding positions (Fig. 3d). Mutating these conserved E389 and R418 in ancDA to the corresponding D and H in ancDADS respectively resulted in a total loss of DA activity (Fig. 3e), indicating that D-to-E and H-to-R substitutions in OCs are evolutionarily indispensable for the emergence of Diels−Alder activity. However, ancDADS-D389E-H418R (ancDADS-mut2a) failed to give any D-A product when incubated with diene **3** and dienophile **4**, indicating that the

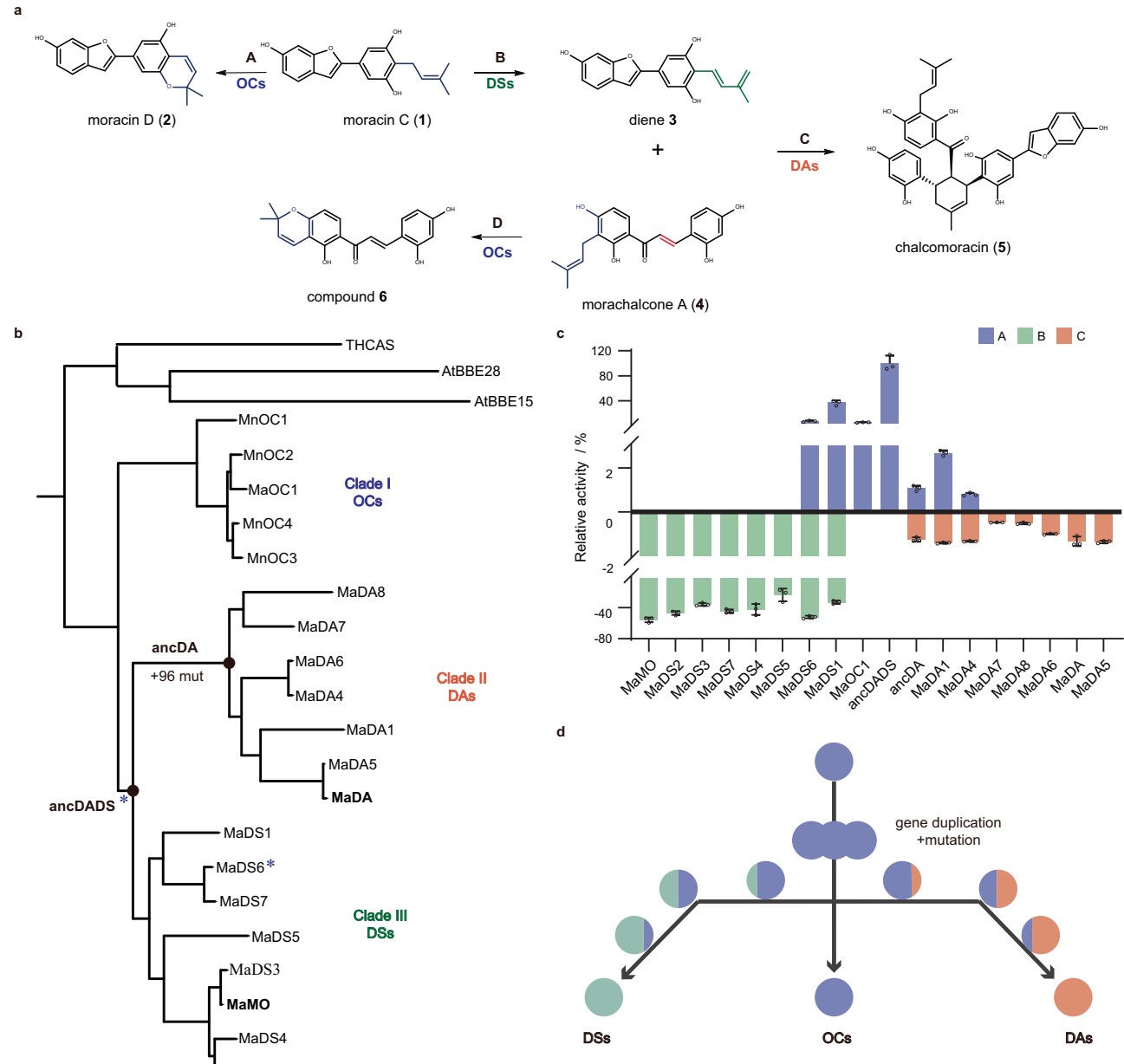

**Fig. 2 | Functional characterization and predicted evolutionary trajectory of DAs, DSs, and OCs. a** The schematic diagram depicts the biosynthetic enzymes of chalcomoracin, moracin D, and **6** in *Morus alba*. OCs catalyze the oxidative cyclization reaction of moracin C (**1**) to form moracin D (**2**) (reaction **A**), while DSs catalyze the oxidative dehydrogenation reaction of moracin C (**1**) to give diene **3** (reaction **B**). The biosynthesis of chalcomoracin occurs through the Diels–Alder reaction between dienophile **4** and diene **3** (reaction **C**). **b** A phylogenetic tree provides insight into the evolutionary relationship between Diels-Alderases (DAs, orange), Diene Synthases (DSs, green), and FAD-dependent oxidocyclases (OCs, light purple) in *Morus alba*. Enzymes labeled with blue asterisks (i.e. ancDADS and MaDSs) can recognize morachalcone A (**4**) as substrate. The abbreviations "Ma" and "Mn" represent *Morus alba* and *Morus notabilis*, respectively. **c** Relative activities of

these three types of enzymes. HPLC analysis was employed to quantify the relative enzymatic activities of OCs (reaction **A**), DSs (reaction **B**), and DAs (reaction **C**). These activities are represented by light purple, turquoise, and orange, respectively. The enzyme assay for detecting D-A activity was conducted with 5 µg of purified protein (0.1–2 mg/mL), 1 µL of diene **3** (100 µM), 1 µL of morachalcone A (**4**) (100 µM) and a 20 mM Tris-HCl solution at pH 8.0. The enzyme assay conditions for detecting oxidative activity included: 5 µg of purified protein, 1 µL of moracin C (**1**) (100 µM), and 20 mM Tris-HCl, pH 8.0. The mixture was then incubated at 50 °C for 7 min. The data are presented as mean values ± standard error (SE), with error bars indicating the standard deviations of three independent measurements. **d** Inferred evolutionary trajectories. Circles of different colors correspond to distinct enzyme types, while circles with two colors indicate bifunctional enzymes.

emergence of the catalytic residues alone is insufficient for ancDADS to acquire the new DA activity (Fig. 4).

To identify other key residues that facilitate the function transformation from OCs to DAs, we examined the distinguishing sequences between these two types of enzymes. Based on the sequence analysis, ancDA evolutionally obtained the DA activity from ancDADS via mutagenesis at 96 differential residues (Supplementary Fig. 7).

Further sequence analysis revealed that there are 21 differential residues, including E389D and R418H between the pockets of ancDA and ancDADS (Supplementary Fig. 6b, c). We postulated that these 21 residues might be sufficient to convert OCs to DAs. To probe this hypothesis, we generated an ancDADS mutant called ancDADS-mut21 by replacing these specific residues with their corresponding ones in ancDA. Enzymatic assays showed that ancDADS-mut21 exhibited the

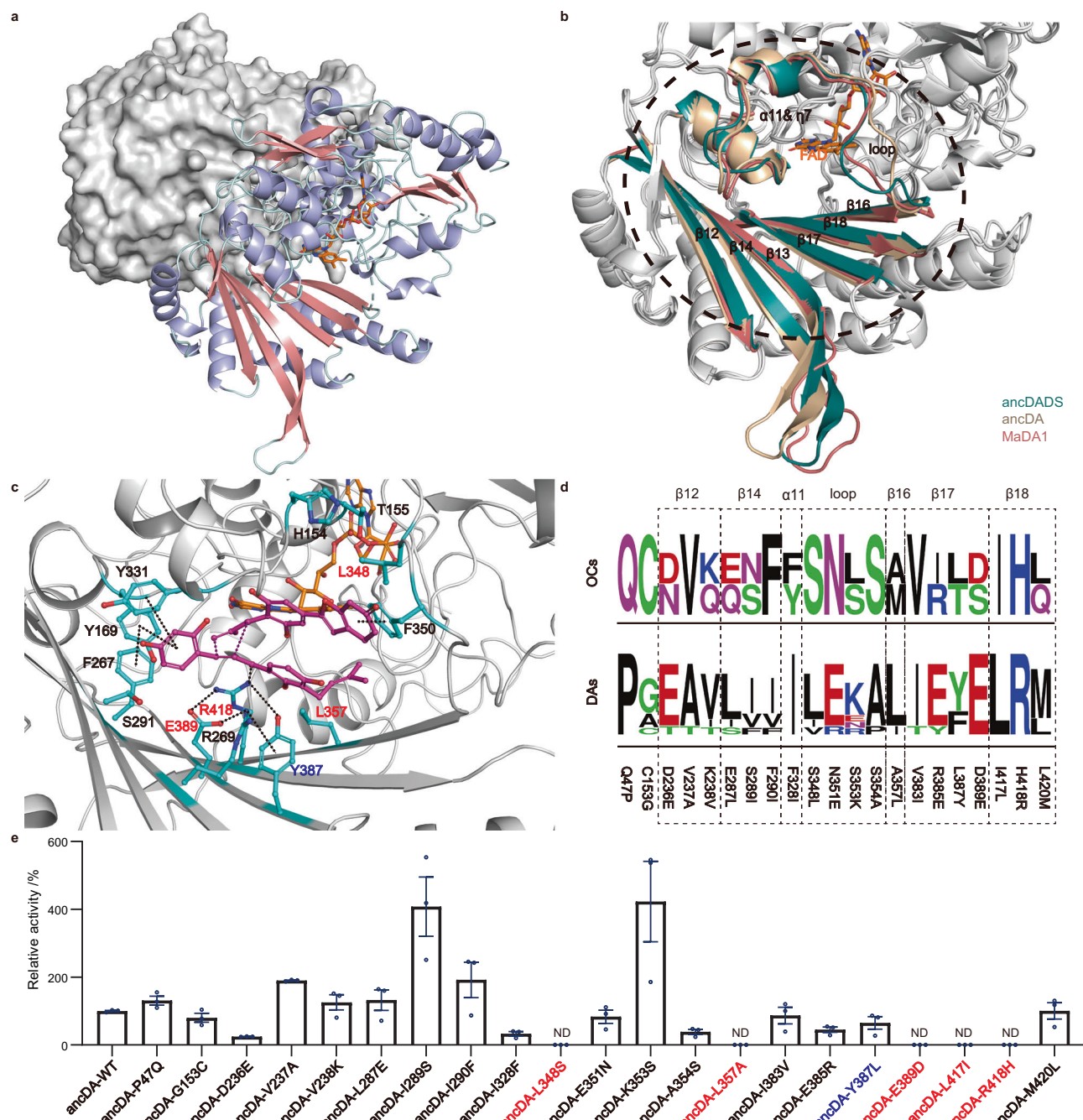

**Fig. 3 | The structure, binding pockets, and key mutant residues of DAs. a** The crystal structure of the MaDA1 (PDB ID: 7YAV) is a dimer. One chain is depicted as a cartoon, while the other appears as a gray surface. The cofactor FAD is shown in orange. **b** Superimposition of the structures of ancDA, ancDADS, and MaDA1. The active pockets of these enzymes are shown in different colors. The structures of ancDA and ancDADS were predicted by AlphaFold. **c** The binding pocket of ancDA. The transition state (*endo*-5) of the substrate appears in purple. The purple dashed line indicates the site of the chemical reaction, while the black dashed line represents the interactions between the substrates and the surrounding residues. The nonconserved residues are highlighted in red or blue font, while the conserved residues are shown in black. **d** Sequence logos illustrate the 21 different residues between DAs and OCs. **e** Diels−Alder activity of ancDA and its mutants, using diene **3** and morachalcone A (**4**) as substrates. The Diels−Alder reaction was conducted in 100 µl of 20 mM Tris-HCl buffer containing 5 µg of the mutants, with 1 µL of dienophiles **4** (100 µM) and 1 µL of dienes **3** (100 µM). The reaction took place at pH 8.0 and 70 °C for 7 min. Relative activity was measured using UPLC and calculated, considering the activity of wild-type and to be 100%. "ND" signifies not detected. The data are presented as mean values ± standard error (SE), with error bars indicating the standard deviations of three independent measurements.

ability to catalyze the D-A reaction between diene **3** and dienophile **4** (Supplementary Fig. 8), confirming that additional substitutions among these 21 residues are required for the emergence of DA activity in BBE-like enzymes.

To find the missing residues for the emergence of DA activity, we then mutated each of the remaining 19 differential residues in ancDA to the corresponding residues from ancDADS. We found that three ancDA variants completely lost their DA activity: ancDA-L348S, ancDA-L357A, and ancDA-L417I (Fig. 3e), indicating that besides E389 and R418, residues L348, L357, and L417 are also important for DA activity, which is also consistent with their high conservation in all DAs (Fig. 3d). Therefore, we concurrently introduced the corresponding five

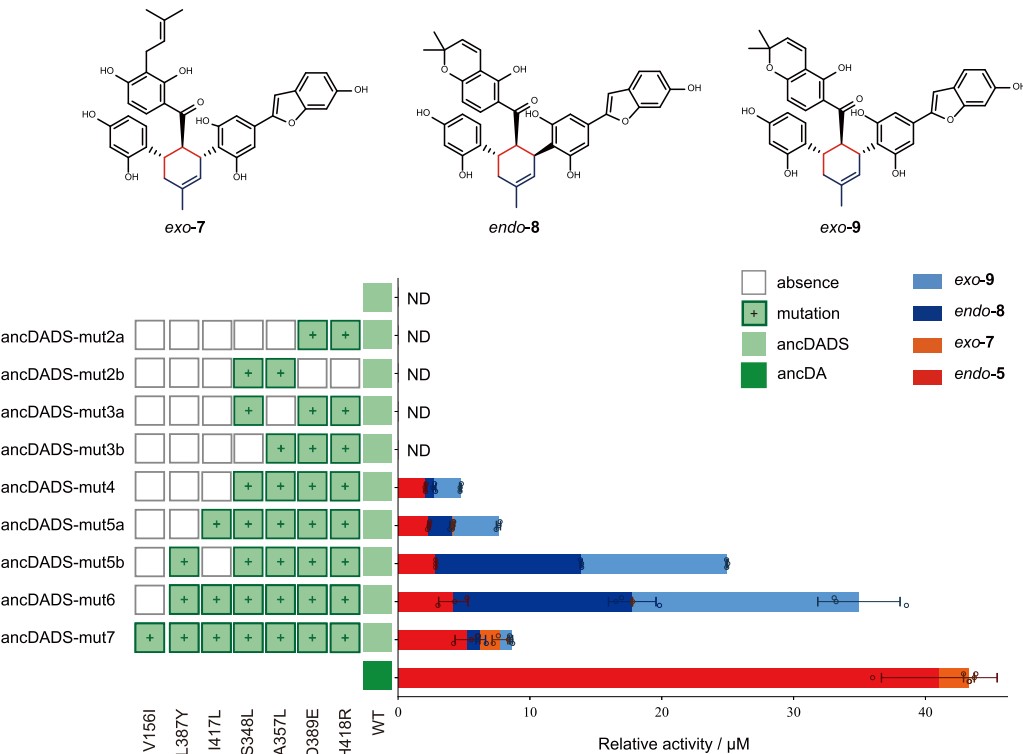

**Fig. 4 | Mutations enabling ancDADS to gain D-A activity.** The D−A products produced by ancDADS mutants were measured in a 100 µL reaction system. This system consisted of 15 µg of mutants, 1 µL of diene **3** (100 µM), 1 µL of dienophile **4** (100 µM), and a 20 mM Tris-HCl buffer (pH 8.0). The reaction was conducted at 50 °C for 2 h. Three sets of replicates were performed for each mutant. The abbreviation "ND" indicates "not detected". The data are presented as mean values ± standard error (SE), with error bars indicating the standard deviations of three independent measurements.

different residues from ancDA to ancDADS, generating an ancDADS variant called ancDADS-mut5a. When ancDADS-mut5a was incubated with diene **3** and morachalcone A (**4**), four D-A adducts, namely chalcomoracin (*endo*-**5**), mongolicin F (*exo*-**7**), and their corresponding oxidative products *endo*-**8** and *exo*-**9**, were observed in this assay (Fig. 4). In addition, ancDADS-mut5a remained its original oxidative cyclization activity by converting dienophile (**4**) into compound **6**, which could further serve as a dienophile of ancDADS-mut5a in the presence of diene **3** to give *endo*-**8** and *exo*-**9** (Supplementary Fig. 8). Interestingly, ancDADS-mut5a could also recognize D-A products chalcomoracin and mongolicin F as its substrate and oxidize them into the cyclized products *endo*-**8** and *exo*-**9**, respectively (Supplementary Fig. 8). The results demonstrated that these five substitutions are sufficient for ancDADS to acquire the DA activity.

Based on our computational study, L348 may weakly interact with the diene moiety in the transition state via hydrophobic interactions (Fig. 3c). Nevertheless, through a site-blocking effect, L348 may impact the localization of F350, which forms a crucial π−π interaction with diene **3**, and thus enhances the binding of diene **3** in the active pocket indirectly. Correspondingly, L357 can affect the localization of R418 through a similar site-blocking effect, pushing R418 closer to the dienophile **4** and thus indirectly affecting the catalytic activity of the DAs. Conversely, no direct interaction between L417 and the transition state or other essential residues was found in the MD simulations. Moreover, further enzymatic assays with longer incubation time also suggested that L417I has a less negative influence on the DA activity of ancDA compared to the other 4 substitutions (e.g., L348S, L357A, E389D, and R418H) (Supplementary Fig. 9). Therefore, L417 is an important but not necessary substitution in the emergence of DA activity. To verify this proposal, we excluded the I417L substitution in ancDADS-mut5a. We constructed a quadruple variant (ancDADS-mut4) to confirm this hypothesis. Enzymatic assays showed that ancDADS-mut4 exhibited a comparatively lower DA activity than ancDADS-mut5a (Supplementary Fig. 8 and Fig. 4), supporting that I417L is not an evolutionarily indispensable substitution but shows a beneficial effect on the evolution of DAs in Moraceae plants.

To further minimize the number of evolutionarily indispensable residues, we created two ancDADS variants with triple substitutions, designated as ancDADS-mut3a (L348S-E389D-R418H) and ancDADS-mut3b (L357A-E389D-R418H). Neither of the two variants produced any D-A product, thereby confirming the critical roles of L348S and L357A substitution in the emergence of DA activity (Fig. 4). Additionally, we constructed an ancDADS variant called ancDADS-mut2b, which only had the L348S and L357A substitutions. No DA activity was detected for this variant, suggesting that these two substitutions alone are necessary but insufficient for the emergence of DA activity in ancDA. Collectively, these results demonstrate that the four substitutions (S348L, A357L, D389E, and H418R) are indispensable for the evolution from ancDADS to ancDA and the acquisition of DA activity.

Apart from the four residues above, residue Y387 may contribute to the enhancement of DA activity in ancDA since it may form hydrogen bonding networks with E389 and R418 (Fig. 3c). Indeed, the introduction of L387Y substitution into ancDADS-mut4 (i.e., ancDADS-mut5b) resulted in a nearly fivefold increase in its DA activity (Fig. 4). Similarly, introducing this substitution into ancDADS-mut5a also resulted in a 4.6-fold increase in the DA activity and the resultant ancDADS-mut6 (containing S348L, A357L, L387Y, D389E, I417L, and H418R substitutions) exhibited a comparable enzymatic activity with that of ancDA (Fig. 4).

On the other hand, we also noticed that the "gatekeeping" residues of the oxygen binding pocket[40] are different between DAs and OCs. Sequence analysis showed a conserved residue V in both OCs and DSs but a bulkier residue I in some DAs (Supplementary Fig. 10). Previous studies have demonstrated that this V to I substitution will

dramatically reduce the oxygen-dependent oxidative activity[41]. Thus, we believe this substitution might be necessary to evolve standalone DAs from their bifunctional ancestors. We further introduced the V156I substitution into the bifunctional ancestor ancDADS-mut6 to verify this hypothesis. As expected, the ancDADS-mut7 showed reduced oxidative cyclization activity, with fewer *endo*-**8** and *exo*-**9** products observed in the enzymatic assay (Fig. 4).

## Substrate-binding specificity was tuned during the evolution of DAs

Molecular dynamics (MD) simulation and experimental analysis were conducted to investigate the substrate-binding pattern change in the evolutionary process from OCs to DAs. Our MD simulation revealed that both morachalcone A (**4**) and moracin C (**1**) formed hydrogen bonding interaction with Y92 and R269 in ancDADS (Supplementary Fig. 11a, b), resulting in a competitive binding mode for these two substrates (Fig. 5a). This conclusion is supported by biochemical evidence showing that morachalcone A (**4**) competitively inhibits the ancDADS-catalyzed oxidative cyclization of moracin C (**1**) with a Ki value of 7.34 μM (Fig. 5b). On the other hand, our MD simulations showed that the transition state for the intermolecular D-A reaction between diene **3** and morachalcone A (**4**) could not maintain a stable interaction with the active site of ancDADS during the simulations (Fig. 5c), which is consistent with our observation that ancDADS is unable to catalyze this intermolecular D–A reaction. Taken together, these results indicate that ancDADS cannot maintain concurrent binding of both diene and dienophile in the appropriate manner required for the intermolecular D-A reaction within its active pocket.

To probe the roles of S348L, A357L, D389E, and H418R substitutions in enabling ancDADS to acquire DA activity, MD simulations of the transition state in complex with ancDADS-mut4 were also performed. In contrast, the transition state maintains an interaction with the active site of ancDADS-mut4 as well as ancDA and MaDA1 over the course of our simulation (Fig. 5c). Moreover, a similar binding model of the transition state in complex with ancDADS-mut4 as ancDA and MaDA1 could be deduced from the MD simulations, wherein the essential substrate-enzyme interactions[30] for maintaining the DA activity including the hydrogen bonding interaction between R418 and morachalcone A (**4**) and the π–π interaction between F350 with diene **3** were also formed (Fig. 5d), indicating that ancDADS-mut4 shares a similar catalytic mechanism for the intermolecular D–A reaction with the ancDA and the extant DAs.

With a more profound understanding of how diene **3** and morachalcone A (**4**) bind simultaneously to ancDADS-mut4, we uncovered the substrate-binding pose changes between ancDADS and ancDADS-mut4. In contrast to ancDADS, morachalcone A (**4**) could form a new hydrogen bonding interaction with R418 and Y169 (Fig. 5d) and thus undergoes a significant swing in ancDADS-mut4 (Supplementary Fig. 11c). Meanwhile, the S348L substitution in ancDADS enables the formation of a new π-π interaction between the benzofuran group of diene **3** and F350 (Fig. 5d). As a result, diene **3** undergoes a counterclockwise rotation in ancDADS-mut4 compared to ancDADS (Supplementary Fig. 11d). Thus, these four identified substitutions (S348L, A357L, D389E, and H418R) collectively facilitate the concurrent binding of the substrates by changing their binding poses, which eventually provides the foundation of new DA activity in OCs.

Based on the results above, we propose an evolutional model regarding the emergence of DAs in Moraceae plants (Fig. 5f). The evolution process began with an OC from the BBE-like enzyme family that acted as a progenitor for the evolution of DAs. This enzyme recognizes both moracin C (**1**) and morachalcone A (**4**). It catalyzes the oxidative cyclization reaction of these two substrates in a competitive way (Fig. 5f). Subsequently, gene duplication events occurred, resulting in multiple copies of this original enzyme that further neofunctionalised through substitutions and natural selection. Four

substitutions, specifically S348L, A357L, D389E, and H418R, altered the binding mode of diene **3** and morachalcone A (**4**) from the competitive accommodation to a simultaneous one in the enzyme, and provided the catalytic residues essential for the DA activity, resulting in the functional transition from OCs into DAs in Moraceae plants.

## Discussion

The evolutionary mechanism of Mulberry DAs, unique natural product biosynthetic enzymes that catalyze intermolecular D–A reactions in plants, was investigated in this study. Through phylogenetic analysis, ASR, and functional characterization, we found that DAs, along with functionally related DSs, have evolved from OCs, leading to the emergence of D–A-type adducts in Moraceae plants. This evolutionary strategy represents a rare case for metabolic pathway development by evolving the same ancestor to two functionally different enzymes catalyzing two consecutive reactions[42–44].

All MaDA homologs were only found in Moraceae species (Supplementary Table 1), consistent with the observation that the natural products of the D-A type adducts were exclusively isolated in Moraceae species[15]. Thus, these intermolecular DAs represent lineage-specific phenotypes that are evolutionarily unique to the Moraceae family. Furthermore, our bioinformatic analysis revealed that DAs, DSs, and OCs typically lack introns and are scattered across the genome (Supplementary Fig. 12a, b). We also discovered that the BBE-like enzyme family encompasses both intronless and intron-rich gene copies (Supplementary Fig. 12c). This leads us to infer that Moraceae DAs may have inherited intronless genes from earlier ancestors before OCs, which may have contributed to their neofunctionalization[45,46]. Meanwhile, we also found that these BBE-like enzymes mainly function in roots and respond differently to UV treatment (Supplementary Fig. 12d, e), indicating the complexity in spatial- and temporal regulation of D-A adducts biosynthesis in Moraceae plants. This observation also suggests that the Moraceae has developed DAs that potentially play a role in responding to abiotic stress, such as UV irradiation. This observation aligns with prior studies indicating the evolution of new genes that confer evolutionary advantages to their host organisms[47,48].

In conclusion, our results shed light on the origin and evolution of the plant-derived intermolecular DAs. This study deepens our understanding of the evolutionary strategy of nature to generate functionally novel enzymes and expand metabolic pathways. Understanding the mechanisms of how DAs naturally evolved will ultimately accelerate the discovery and rational design of new DAs that are potentially useful in producing structurally diverse D−A-type bioactive molecules.

## Methods
### Sequence mining of BBE-like enzymes and DAs
Protein sequences in the nonredundant protein database (nr) and UniProt database[49] were searched using HMMER v3.3.2[50], respectively, from which 8,123 and 82 protein sequences containing both FAD_binding_4 and BBE domains were extracted respectively. These proteins were renamed according to their annotations. Additionally, 55 protein sequences of BBE-like enzymes were reported by ref. 19 were extracted. After removing redundant sequences, a total of 94 BBE-like enzymes were obtained. To mine putative DAs, initial data mining was performed based on the known databases using the *Morus alba* Diels-Alderase MaDA (GenBank: QIB03073.1) and MaDA1 (gene bank: UJH93677.1) as a query. The homologs of MaDA in the nonredundant protein database (nr) and UniProtKB/Swiss-Prot database of NCBI[51] were retrieved with an E value threshold of 0.05 The putative DAs identified by phylogenetic tree analysis all come from Moraceae plants. To determine more DAs in Moraceae plants, publicly available raw sequencing data of Moraceae transcriptomes were de novo assembled (see Supplementary Table 1 for more information). Sequence quality was examined by using FastQC, and low-quality reads were removed. Trinity[52] was employed for the de novo assembly of the

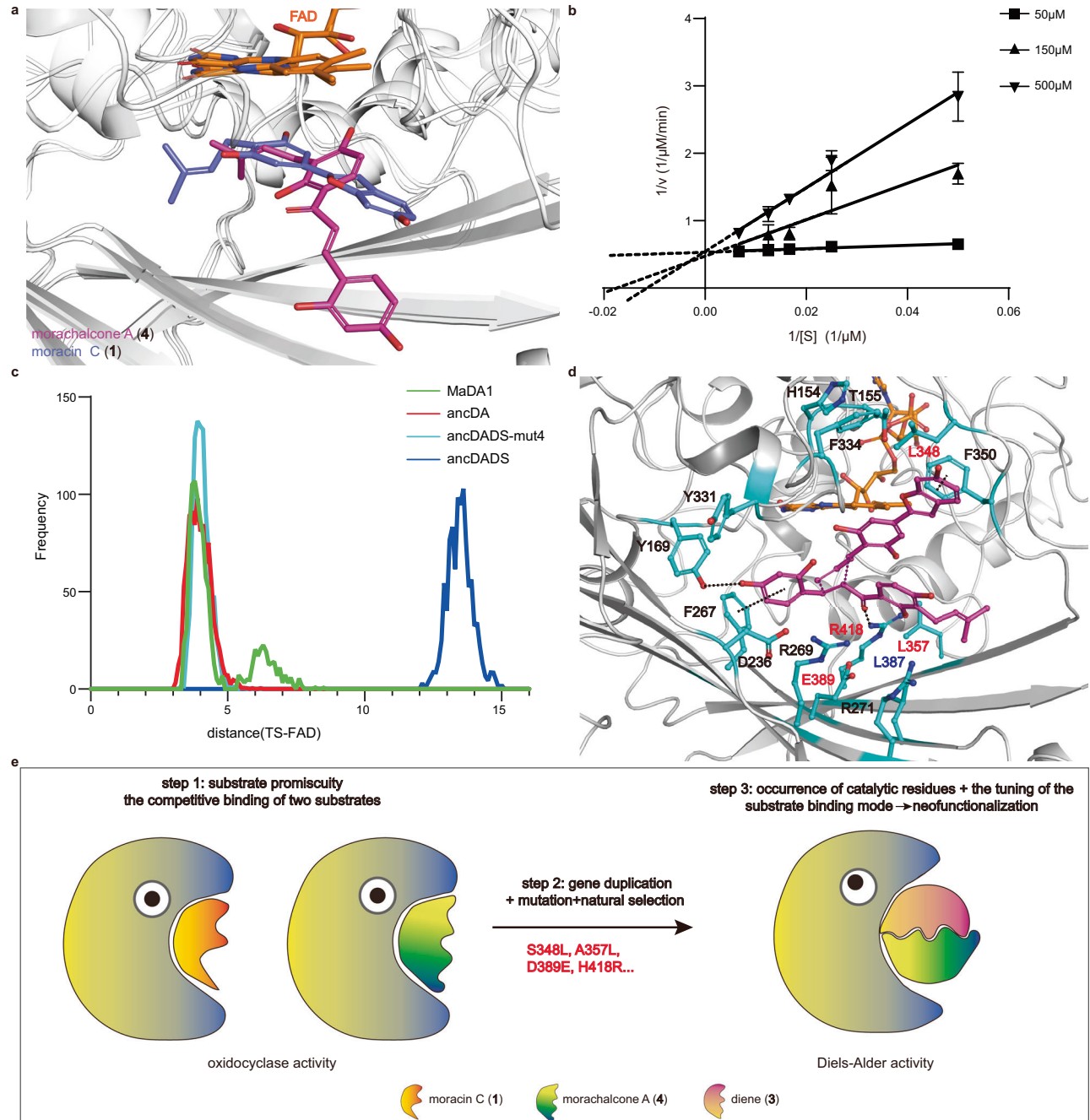

**Fig. 5 | The tuning of the substrate-binding mode. a** Overlay structures of ancDADS with moracin C (**1**) in blue and morachalcone A (**4**) in purple. **b** A Lineweaver–Burk plot showing competitive inhibition of the ancDADS-catalyzed oxidative cyclization reaction of moracin C (**1**) by dienophile **4**. 100 μL enzyme reaction system: 1.5 μL of moracin C (20–150 μM), 0.3 μg of ancDADS, 50/150/ 500 μM of morachalcone A (**4**), 20 mM Tris-HCl buffer (pH 8.0). The data were presented as mean values ± standard error (SE), with error bars indicating the standard deviations of three independent measurements. **c** Frequency distribution histogram of the distance between FAD and the *endo*-transition state (TS). **d** The binding model of ancDADS-mut4-TS deduced from 500 ns MD simulations of the TS in complex with the enzyme. A black dashed line represents interactions between the substrate and surrounding residues. Nonconserved residues are indicated using red and blue fonts, while conserved residues are shown in black. **e** Evolution models of DAs from OCs.

cleaned FASTQ sequences. TransDecoder v5.5.0 was used to identify CDS regions after contig assembly with TGICL[53]. Then, HMMER v3.3.2[50] was used to identify BBE-like enzymes. The genes with a FAD-binding domain (PF01565) and the BBE domain (PF08031) were retrieved by HMMER. The phylogenetic tree was used to identify the DA clade.

**Phylogenetic analysis and ancestral sequence reconstruction**
For the phylogenetic tree analysis of BBE-like enzymes, three previously reported genes (AtCKK, ZmCKX1, and VAO) in the FAD-linked oxidase

family are closely related to the BBE-like enzyme family were included as outgroups. The multiple sequence alignment of BBE-like enzymes was generated using the program MAFFT v7[54]. The Database of Aligned Structural Homologs (DASH) was used to add homolog structures. The E-INS-i iterative refinement method was chosen for the comparison of the sequences. Redundant sequences were removed by using CD-HIT[55] with -c 0.8. Gblock[56] was then used to extract conserved regions with a maximum number of contiguous nonconserved positions of 50, allowing for gap positions. Any poorly matched sequences were also manually

removed. DAMBE v5[57] was used to detect sequence replacement saturation. A maximum likelihood (ML) phylogenetic tree was inferred with IQTREE v1.6.12[58] with the following parameters: -m MFP -bb 1000 -bnni -redo. The best-fitting model chosen according to BIC for simplified (Fig. 1) and full phylogenetic trees (Supplementary Fig.1) are WAG + F + R6 and WAG + F + R10, respectively. For the phylogenetic tree of DAs, DSs, and OCs, the best-fitting model found by IQTREE was JTT + R4. Bayesian inference-based phylogenetic trees were also constructed. The Bayesian inference-based method was implemented using MrBayes 3.2.7a[59] to construct the phylogenetic tree. The parameters of MrBayes were as follows: lset Nucmodel=Protein Nst=Mixed rates=Invgamma; mcmcp ngen=1000000 samplefreq=1000 diagnfreq=1000. For candidate genes from other species, the protein sequences with fewer than 450 or more than 650 amino acids were removed in order to enhance the accuracy of the phylogenetic tree. ASR was performed with codeML in PAML v4[60] using the default codon settings. The ancestral sequence of DAs predicted by codeML was named ancDA, and the common ancestor of DAs and DSs was named ancDADS. The signal peptide in the N-terminal amino acids was predicted with significant ambiguity, and identical adapter sequences derived from MaDA and MaMO (GenBank: QIB03072.1) were therefore added to ancDA and ancDADS, respectively. Ambiguously aligned positions were handled manually based on the extant sequences. Ultimately, we obtained ancDA and ancDADS, containing 526 amino acids.

### *Morus alba* RNA preparation and sequencing

*Morus alba* leaves were collected from Peking University, Beijing, China. The collected samples were immediately frozen in liquid nitrogen and stored at −80 °C until being used for RNA extraction. The total RNA of *Morus alba* was isolated from the frozen leaves by using TRIzol (Invitrogen, Carlsbad, CA, USA) following the manufacturer's protocols and was reverse transcribed using a SMARTer RACE cDNA amplification kit (Clontech Inc.). According to the manufacturer's protocol, the reverse-transcribed products were subjected to the rapid amplification of cDNA ends (RACE). cDNA was used to amplify the predicted genes. For transcriptional analysis, the leaves were pretreated with UV light for 1 h in a dish containing water and then collected for transcriptional sequencing after incubation in the dish for another 0, 1, 4, 13, 24, and 48 h.

### Gene cloning, site-directed mutation, protein expression, and purification

The genes encoding putative DAs, DSs, and OCs were amplified from the cDNA of *Morus alba* by PCR using FastPfu DNA polymerase (TransGen Biotech, China). The sequences of the primers used in this study are listed in Supplementary Table 2. The ancestral genes ancDADS and ancDA were obtained from Xiang Hong Company by gene synthesis. The amplified DNAs were recombined in the pISUMOstar insect intracellular vector. The Bac-to-Bac baculovirus expression system was used for protein expression. Recombinant baculoviruses were generated and amplified in Sf9 insect cells cultured in SIM SF medium (Sino Biological Inc.). The recombinant gene was expressed as a secreted protein in sf9 cells with an N-terminal 6 × His-SUMO fusion tag. Sf9 cells were infected with the recombinant virus (multiplicity of infection of 2) at a $1.5–2.0 × 10^6$ cells ml$^{-1}$ density. After 48 h of infection, the medium was harvested and concentrated using a Hydrosart Ultrafilter (Sartorius). The concentrated medium was then exchanged into the binding buffer, consisting of 20 mM Tris-HCl, pH 8.0, and 200 mM NaCl. Subsequently, the protein was purified using Ni-NTA resin and stored at 4 °C. SDS−polyacrylamide gel electrophoresis (SDS−PAGE) was used to check the purity of the eluted protein. The proteins were stored at −4 °C for activity assays or flash-frozen in liquid nitrogen and stored at −80 °C for long-term use.

### Activity assays

Enzymatic assays of DA activity were carried out in a reaction mixture (100 μL) that contained the following components: 5 μg purified protein (0.1–2 mg/mL), 1 μL morachalcone A (**4**) (100 μM), 1.0 μL diene **3** (100 μM) which was in situ generated according to the reported procedure[18], and 20 mM Tris-HCl, pH 8.0. The reaction mixture was incubated at 50 °C for 7 min. The assays were quenched by adding 150 μL ice-cold methanol and centrifuged at 15,000×*g* for 10 min to spin down enzymes and debris. The products produced by each enzyme were analysed and quantified by ACQUITY ultra-performance liquid chromatography (UPLC) H-class system (Waters) integrated with an SQ Detector 2 that features an electrospray ionization source. The ACQUITY UPLC BEH C18 column employed for general UPLC/mass spectrometry (MS) analysis (50 mm length, 2.1 mm inner diameter, 1.7-μm particle size; Waters) facilitated the flow at 0.3 ml/min at an operational temperature of 40 °C. A gradient elution consisting of water (A) and acetonitrile (MeCN) (B) was applied, starting with 70% A and 30% B for the first 3.5 min of the prerun, transitioning to 98% A and 2% B over 5 min, and concluding with 100% B in the final 2 min for elution. Their relative activities were calculated by comparing their peak areas of products. Three parallel assays were carried out for each enzyme. The enzymatic assays of oxidative cyclization activity were carried out and analyzed according to a similar procedure to that described above, in which 1 μL moracin C (**1**) or morachalcone A (**4**) (final concentration, 10 μM) was used as the substrate instead of morachalcone A (**4**) and diene **3**.

### Crystallization of MaDA1, data collection, and structure determination

MaDA1 was digested with tobacco etch virus protease to prepare the protein sample for crystallization to remove the N-terminal 6× His-SUMO fusion tag. The tag and the tobacco etch virus protease were separated with MaDA1 using size exclusion chromatography (SEC) on a Superdex 200 10/300 GL column (GE Healthcare). Untagged MaDA1 was purified by cation exchange chromatography using a HiTrap Q column (GE Healthcare), followed by SEC on a Superdex 200 10/300 GL column. The proteins were concentrated at approximately 10 mg/mL. Crystals were grown using a sitting drop vapor diffusion method at 18 °C. MaDA1 was crystallized in 0.1 M Bis-Tris (pH 5.5) and 25% PEG3350. The crystals were cryoprotected in crystallization solution supplemented with 30% ethylene glycol for MaDA1 and substantially flash-frozen in liquid nitrogen. X-ray diffraction data were collected at the Shanghai Synchrotron Radiation Facility (beamline BL19U1 for MaDA1) with an X-ray wavelength of 0.97915 Å. The data were processed using HKL2000. The structure was solved via a molecular replacement method using Phaser, with the MaDA3 structure (PDB ID 7E2V) as the search model. The model was built with Coot and refined using Phenix.refine. Ramachandran plot analysis showed that 96.70% of all the residues of MaDA1 are Ramachandran favored, and there are no Ramachandran outliers in the models. Data collection and structure refinement statistics are summarized in Supplementary Table 3.

### Molecular docking and molecular dynamics simulation

The structures of ancDADS, ancDA, and ancDS were calculated using AlphaFold v2.2.0[61] with the following parameters: -m model_1, model_2, model_3, model_4, model_5, and -t 2019-09-21. For the molecular dynamics (MD) simulations, each docking complex featuring the transition state (TS) with ancDA, ancDADS, and ancDADS-mut4 served as the foundational structure. Protonation states were determined using the Protein Preparation Wizard in Maestro. We employed the OPLS4 force field parameter set for both the protein and ligand, accompanied by the TIP3P model for water. Each protein complex was immersed in a pre-equilibrated orthorhombic box with a 10 Å buffer of TIP3P water molecules, and the volume was minimized. Explicit

counterions (Na$^+$ or Cl$^-$) were added to neutralize the systems, and 0.15 M NaCl was introduced to simulate ionic effects in a realistic environment. Molecular dynamics simulations were conducted using the Desmond software package, employing a standard NPT (isothermal-isobaric ensemble) relaxation protocol. The simulation protocol unfolded in multiple stages to explore the system's behavior under diverse conditions. All time units are reported in picoseconds (ps), and energy is expressed in kilocalories per mole (kcal/mol). Utilizing default parameters, the simulation was performed with a single CPU, employing a cutoff radius of 9.0 Å, a time step of [0.002 0.002 0.006] ps, and a total simulation time of 500,000 ps. For temperature control, the Berendsen algorithm was employed with a target temperature of 343.0 K and a coupling time of 1.0 ps. Pressure control was achieved using the NPT ensemble with a pressure setting of 1.01325 bar. The pressure coupling time was 2.0 ps, and the temperature coupling time was 1.0 ps. The restraint protocol employed a force constant of 50 kcal/mol/Å$^2$ on heavy atoms. To define the transition state geometry, a harmonic constraint with a force constant of 500 kcal mol$^{-1}$ Å$^{-2}$ was applied. Torsional constraints were applied to the planes of diene and dienophile carbons with a force constant of 500 kcal mol$^{-1}$ Å$^{-2}$. These restraint parameters were selected to ensure the preservation of the desired structural features during the simulation while allowing for the exploration of dynamic behavior. The radial distribution functions (RDF) between the cofactor FAD and the substrate were calculated using Maestro Version 13.1 from Schrodinger with a range of 300–500 ns. We used Schrodinger's built-in trj_cluster.py script for clustering the trajectories, where -rmsd-asl is all and -s is 500:1000:5. The results were plotted using GraphPad Prism 8. PyMOL[62] generated interaction plots of the enzymes and substrates. The enzyme's binding pocket volume was calculated using POVME 3.0[63]. The spatial coordinates of the side chain of R418, located at the corresponding position in ancDADS, were used as a fixed point to define the pocket. The pocket radius was set to 12 Å with L420 as the seed sphere, and the remaining parameters were set to default. The volume of the pocket was calculated after that.

**Chromosomal location, exon–intron structure analysis, and expression analysis**

Information on the chromosomal location of the identified genes, DAs, DSs, and OCs, in *Morus alba* was obtained from the National Center for Biotechnology Information (NCBI). The chromosomal location of the genes was visually represented using the R package RIdeogram[64] and ggplot2, following their genome annotation position. MScanX[65] was employed to perform a synteny analysis using the all-vs-all blast results. The gene structure annotation of BBE-like enzymes was extracted from GFF files downloaded from NCBI. The Gene Structure Display Server 2.0[66] was utilized to display the genes' exon/intron organization. RNA-seq data of three different tissues of *Morus alba* were obtained from NCBI under the project PRJNA660559. Trimmomatic v0.39[67] was utilized for the cleaning of reads. Kallisto software[68] was used to measure the TPM values. The expression pattern was displayed using the R package pheatmap.

**Reporting summary**

Further information on research design is available in the Nature Portfolio Reporting Summary linked to this article.

## Data availability

The data supporting this study's findings are available with this article and its Supplementary Information or from the corresponding authors upon request. The origins of the raw sequencing data analyzed in this study are elaborated in Supplementary Table 1. The gene sequences of MaDA5-8, MaDS1-7, and MaOC1 are deposited in GenBank under accession NO.ON745422, ON787914, ON787915, ON787916, ON787917, ON787918, ON787919, ON787920, ON787921, ON787922, ON787923, ON787924. The structural factors and coordinates of MaDA1 are deposited in the Protein Data Bank under ID 7YAV. Source data are provided in this paper. Source data are provided with this paper.

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

## Acknowledgements

The data analysis was performed on the High-performance Computing Platform of Peking University. This work is supported by the National Key Research & Development Plan (2021YFC2102900 to L.G.; 2022YFC3401500 and 2022YFC2502500 to X.L.), the National Natural Science Foundation of China (22193073 and 92253305 to X.L.; 22101009 and 22322701 to L.G.; 22177006 to J.F.), and Beijing National Laboratory for Molecular Sciences (BNLMS-CXX-202106 to X.L.). X.L. is supported by the New Cornerstone Science Foundation through the XPLORER PRIZE.

## Author contributions

X.L. conceived the original research plans; X.L. and L.G. managed the whole project; Q.D., N.G., L.G. and X.L. designed the experiments; Q.D. performed the bioinformatic analysis; Q.D. and N.G. conducted the biological and biochemical experiments with the help of L.G. and J.Y.; J.Y., Q.D. and N.G. performed the crystal structure study under the guidance of J.F.; D.W. synthesized small molecules under the guidance of X.L. and L.G.; M.M. conducted anti-microbial screens under the guidance of J.W.; Q.D., N.G., L.G. and X.L.wrote the manuscript with inputs from all the authors.

## Competing interests

The authors declare no competing interests.
