## [Peer Review File · Nature Communications]

The evolutionary origin of naturally occurring intermolecular Diels-Alderases from *Morus alba*REVIEWER COMMENTS

Reviewer #1 (Remarks to the Author):

The manuscript (Manuscript ID: NCOMMS-23-56002-T) by Xiaoguang Lei, et al. entitled "The evolutionary origin of naturally occurring intermolecular Diels-Alderases from *Morus alba*" explores evolutionary origins of the intermolecular Diels-Alderases in the biosynthesis of plant-derived Diels-Alder type secondary metabolites. The authors claim the findings suggest that these Diels-Alderases have evolved from an ancestor functioning as a flavin adenine dinucleotide dependent oxidocyclase, which catalyses the oxidative cyclisation reactions of isoprenoid substituted phenolic compounds. Through crystal structure determination, computational calculations, and site-directed mutagenesis experiments, they identified several critical mutations, including S348L, A357L, D389E, and H418R, that alter the substrate-binding mode and enable the flavin adenine dinucleotide dependent oxidocyclases to gain intermolecular Diels-Alderase activity during evolution. The results provide mechanistic insights into the evolutionary rationale of Diels-Alderases and pave the way for mining and engineering new Diels-Alderases from other protein families. However, MaDA could not catalyze stereoselective Diels-Alder reaction during the formation of chalomoracin (5) as an endo form and 7 as an exo form catalyzed by wild type MaDA in Figure 4. The authors should show the productions of both compounds by using a knock-out strain of the gene, MaDA, in *Morus alba*. Regarding the non-stereoselective Diels-Alder reaction, the authors should clearly show the transition state energies for both transformations instead of Figure 5c, which is hard to understand what you show. They have to show the computational results such as Figure 4b in reference number 17 in this manuscript or Figure 4b in Sato, M., et al. *Nature Catalysis*, 2021, 4, 223-232. The section of antifungal and antibacterial activity of Diels-Alder-type adducts should be deleted in this manuscript because it does not matter with Diels-Alderase. The authors should focus on demonstrating the intermolecular Diels-Alder reaction by MaDA.

Reviewer #2 (Remarks to the Author):

This study seeks to learn how diels-alderases (DAs) in Moraceae plants evolved their mechanisms. Evolutionary mechanisms showed these enzymes emerged from an ancestor with oxidocyclase activity. The authors use phylogenetics, enzymatic assays, structural studies, MD simulations and mutagenesis to study ancestral and modern variants to identify amino acids that drove the acquisition of diels-Alderase activity and substrate binding. These studies are rigorous and provide mechanistic insight into how this family of enzymes evolved, while also giving insight into how proteins can be rationally engineered to achieve new catalytic functions.

A few comments are provided below to help to improve the manuscript. After these comments are addressed, acceptance is recommended.

This is a semantic matter: authors should consider replacing the word mutations/mutagenesis in connection to evolutionary changes with 'substitution'.

Authors state that the ionic hydrogen bond involving E389 and R418 promotes the D-A reaction by lowering the LUMO energy. It is not clear how this determination is made, as no energy calculation is reported in the manuscript. The authors should provide an explanation of this result or a reference explaining how this conclusion was arrived at.

Authors have used MD simulations to determine that substrates or transitions do or do not bind to enzymes. However, the only data they show to support this claim is distances, which is not enough. Distances can be used to support a more sustained interaction between components of the protein-TS complex, but they cannot be used to describe tight binding (or even binding), unless binding affinities have been calculated, which can be done from MD simulations. Authors should either perform binding calculations or remove the quantitative language about binding, replacing it with something like 'transition state maintains an interaction with the active site over the course of our simulation'. This comment refers to data shown in Figures 5C and 5D.

Because models were prepared by AlphaFold, authors should consider providing the confidence values (per-residue) of their structures. This will provide a sense of how reliable the models are.

Critical details of the MD simulation are missing. Was solvent added, and how much? Were ions added? Were minimization and equilibration performed? What type of ensemble, thermostat etc were used.

Reviewer #3 (Remarks to the Author):

Lei et al. report an interesting study to trace the evolutionary origin of the intermolecular Diels-Alderase and their functionally coupled dehydrogenases in *Morus alba* (named MaDAs and MaDSs, respectively). Based on a thorough phylogenetic analysis with an association of enzymatic activity assays, the authors raised the hypotheses that both MaDAs and MaDSs share with the oxidative cyclases in Moraceae plants (named MaOCs) an ancestor enzyme possessing a berberine bridge enzyme (BBE)-like fold and that OC activity appears to be more ancestral and can be developed to individual dedicated OCs, DAs and DSs through gene duplication and mutagenesis. To validate these hypotheses, they constructed the ancestral Diels-Alderase (ancDA) and the last common ancestor of DAs and DSs (ancDADS) for comparative analysis. Consequently, the ancDADS, which basically is an OC and has no DA and DS activities, was developed to an enzyme with DA activity via a minimum of the mutations of four key residues. Eventually, computational studies rationalized the catalytic processes for ancDA, ancDADS and ancDADS-derived enzymes with DA activity, leading to a model for the evolution of OC to DA.

Overall, this is a very interesting and comprehensive study that showcases how enzymes evolve for functional differentiation to process structurally related substrates in different ways by combining genetic, biochemical, structural biology and computational approaches together. The manuscript is well-written. The experiments are clearly and comprehensibly presented, and figures are well-arranged. Understanding the evolutionary origin and mechanism of MaDAs will ultimately facilitate design and develop new enzyme tools for synthetic biology utilization. This reviewer is supportive of publication (although minor revision appears to be necessary) because the results should be of great interest to the interdisciplinary readership of the journal.

Enzymatic activity assays of selected MaOCs, MaDAs and MaDSs revealed that a few are functionally promiscuous, i.e., two DAs and two DSs have OC activity for moracin D formation. Did the authors determine whether DAs have DS activity or DSs have DA activity? If not, could you provide an explanation, particularly based on the evolutionary model shown Fig. 2?

Page 7. The single mutation of I417L in ancDA resulted in the complete loss of DA activity. This results appears to be a bit contradictory to the weak DA activity of ancDADS-mut4 in which the I417L mutation is not involved.

Extended Data Fig. 7, the lane for the incubation of ancDADS-mut5 with substrates 3 and 4. The product profile is complicated and have some peaks in addition to those indicated. What do these peaks represent? It seems additional activities to OC and DA activities were observed?

Chemically, the evolution of an OC from a DS is possible, given the notion that cyclization activity can be developed for 1,4-addition by following dehydrogenation immediately. Thus, the mechanism by which MaOCs function is interesting: during the catalysis, does cyclization occur prior to dehydrogenation?

Others:

1. Page 2, abstract, line 35: "plant-derived Diels-Alder-type secondary metabolites". It could be better to specialize to "moraceae plant-derived".
2. Page 2, abstract, line 40: please remove the comma in "D389E, and H418R" and in "and H418R, that alter".
3. Page 3, line 83: please remove the first "typically" in "Unlike most BBE-like enzymes typically, which typically catalyse".

4. Page 4, line 100: "FAD-dependent DAs in Moraceae plants evolve from the BBE-like enzymes" could be better.
5. Page 6, line 177: please leave a blank place between "MaDA1(Extendend Data Fig. 5a".
6. Page 7, line 207: please leave a blank place between "mongolicin F(exo-7)".
7. Page 10, Line 318: the sentence "" needs to be rephrased. It could be better "This evolutionary strategy represents a rare case for metabolic pathway development, by evolving a same ancestor to two new enzymes catalysing two consecutive reactions".

Reviewer #4 (Remarks to the Author):

Although many enzyme-catalyzed Diels-Alder reactions have been identified, the evolutionary mechanisms of natural Diels-Alder enzymes remain unclear. Lei and co-authors use phylogenetic analysis, ancestral sequence reconstruction, and other methods to elucidate the evolutionary origin of Diels-Alder enzymes in Moraceae plants. They found that these Diels-Alder enzymes originated from gene duplication followed by neofunctionalization of BBE-like enzymes with oxidocyclase activity. The results are interesting and novel to the natural product biosynthesis research.

Questions and suggestions:

1. This work involved docking and Molecular dynamics simulation analyses to investigate the substrate binding pattern change in the evolutionary process from OCs to DAs. It might be better to perform quasi-classical direct-dynamics simulations as Ken Houk and co-authors did (J Am Chem Soc, 2016, 138, 3631-3634).
2. In line 70 of page 3, the authors wrote that "the monofunctional DAs PyrE3, SdnG and SpnF". However, the work of Hung-wen Liu and Ken Houk showed that SpnF is a multifunctional enzyme with [4+2], [6+4], and cope rearrangement activities. (J Am Chem Soc, 2016, 138, 3631-3634; PNAS, 2017, 114, 10408-10413.)
3. In line 360 of page 11, the authors wrote, "Protein sequences with fewer than 450 or more than 650 amino acids were removed." Would this prevent the identification of new Diels-Alder enzymes?
4. In line 430 of page, "0. 0.97915 Å" should be replaced by "0.97915 Å".

Point-by-Point Responses

Contents

Response to Reviewer #1	Page 2–8
Response to Reviewer #2	Page 9–15
Response to Reviewer #3	Page 16–24
Response to Reviewer #4	Page 25–29

Response to Reviewer #1

Reviewer #1 (Remarks to the Author):

Comments: The manuscript (Manuscript ID: NCOMMS-23-56002-T) by Xiaoguang Lei et al. entitled “The evolutionary origin of naturally occurring intermolecular Diels-Alderase from *Morus alba*” explores the evolutionary origins of the intermolecular Diels-Alderase in the biosynthesis of plant-derived Diels-Alder type secondary metabolites. The authors claim the findings suggest that these Diels-Alderase have evolved from an ancestor functioning as a flavin adenine dinucleotide dependent oxidocyclase, which catalyses the oxidative cyclisation reactions of isoprenoid substituted phenolic compounds. Through crystal structure determination, computational calculations, and site-directed mutagenesis experiments, they identified several critical mutations, including S348L, A357L, D389E, and H418R, that alter the substrate-binding mode and enable the flavin adenine dinucleotide-dependent oxidocyclases to gain intermolecular Diels-Alderase activity during evolution. The results provide mechanistic insights into the evolutionary rationale of Diels-Alderase and pave the way for mining and engineering new Diels-Alderase from other protein families.

We would like to thank this reviewer for his/her thorough review of the manuscript and for providing valuable feedback.

1. However, MaDA could not catalyze stereoselective Diels-Alder reaction during chalconoracine (5) formation as an endo form and 7 as an exo form catalyzed by wild-type MaDA in Figure 4.

Response #1-1:

As our previous investigations, *Morus alba* Diels-Alderase (MaDAs) are classified into two types: *endo*-selective DAases, such as MaDA and MaDA1, which produce *endo*-configured D-A products as the only product, and *exo*-selected DAases, such as MaDA2 and MaDA3, catalyzing intermolecular D-A reactions with *exo*-selectivity (Gao, L. et al. Nat. Catal. **2021**,4,1059). We also found that residues R294 and R443 are pivotal in regulating *endo/exo* selectivity. In *endo*-selective DAase MaDA, R443 activates the dienophile through a hydrogen bonding interaction, reducing the energy gap between the highest occupied molecular orbital (HOMO) of the diene and the lowest unoccupied molecular orbital (LUMO) of the dienophile (Gao,

L. et al. Nat. Catal. **2021**, 4, 1059). As for *exo*-selected MaDA3, although it also contains the catalytic R443, its major product is *exo*-configured D-A products instead of *endo*-configured ones. This difference was attributed to the existence of a unique and conserved R294 in *exo*-selective MaDA3, which engages in cation- π interaction with the benzene ring of the dienophile, thereby stabilizing the dienophile through electrostatic and dispersion interactions and enhancing enzymatic activity to generate *exo*-products (Gao, L. et al. Nat. Catal. **2021**, 4, 1059).

In this study, we have functionally characterized 5 more extant MaDAs and their ancestor ancDA and found that the R443 is conserved in all these MaDAs, including ancDA. At the same time, MaDSs/MaOCs contain a conserved His at the corresponding position instead (**Fig. R1a**). On the other hand, we also observed that newly identified *exo*-selective DAases MaDA7 and MaDA8 share a conserved R294 as MaDA3. At the same time, the *endo*-selective MaDAs (MaDA, MaDA1, MaDA4-6) have a Gly at the corresponding position (**Fig. R1b**). These results are highly consistent with our previous conclusion that residues R443 and R294 are responsible for the *endo*- and *exo*-selective D-A reactions, respectively.

Figure R1. Statistical analysis of R443 and R294 in MaDAs, MaDSs and MaOCs.

a, Statistical analysis was conducted on R443 within MaDAs, MaDSs and MaOCs. The figure illustrates a phylogenetic tree encompassing MaDAs (Diels-Alderases), MaDSs (diene-synthases), and MaOCs (oxidocyclases), wherein residues corresponding to the catalytic residue R443 are annotated within parentheses. The corresponding residue at positions aligning with R443 in each enzyme was indicated by the single letter codes of amino acids. **b**, The statistical analysis of R294 within MaDAs, MaDSs and MaOCs.

Like the *exo*-selective MaDAs (MaDA7 and MaDA8), ancDA was found to harbor both R294 and R443 (**Fig. R1b**). We believe that this feature of ancDA supports its capacity to catalyze the formation of *exo*-configured D-A products as we observed in Figure 4. Unlike ancDA, the evolutionally more distant ancDADS only contains the conserved R294 in *exo*-selective D-Aases but not the conserved R443 across both *endo*- and *exo*-selective D-Aases. When we introduce the conserved R443 as well as the other three residue into ancDADS, the generated ancDADS-mut4 was proved to catalyse both *endo*- and *exo*-selective Diels-Alder reactions without obvious bias (Figure 4). These results suggested that the evolution of Diels-Alderases (DAases) underwent a process of functional specialization. This transition involved a shift from the ancestral state of enzymes with a broad spectrum of catalytic activities, encompassing both *endo*-type and *exo*-type products, to enzymes exhibiting a tendency toward either *endo* or *exo* products. This aligns with findings reported by some researchers (**Fig. R2**) (Tokuriki, N. et al. Science **2009**, 324, 203 & Khersonsky, O. et al. Annu. Rev. Biochem. **2010**, 79, 471).

Figure R2. Evolutionary model of MaDAs. Under natural selection, OCs acquired a weak DA activity to catalyse the D-A product formation without obvious *endo/exo* selectivity. Then they have gradually enhanced the *endo/exo* selectivity during the evolution. By the end of this process, which typically requires many generations of mutation and selection, the catalytic functions of DAs have been re-shaped into either *endo*-selective or *exo*-selective DAs as we observed in *Morus alba*.

2. The authors should show the productions of both compounds by using a knock-out strain of the gene, MaDA, in *Morus alba*.

Response #1-2:

We would appreciate the excellent suggestion from this reviewer. However, we believe that the gene knockout experiments in *Morus alba* might be too technically challenging to achieve. Although genome editing technologies such as CRISPR-associated protein 9 (Cas9) genome editing systems emerge as robust tools for precise gene modifications, their application encounters significant challenges. These challenges encompass the imperative to develop efficient methodologies for delivering CRISPR and other editing tools to plants and the need for more effective strategies concerning sequence knock-ins and replacements (Mao, Y. et al. Natl. Sci. Rev. **2019**, 6, 421). This complexity is particularly salient when dealing with non-model plants like *Morus alba*. Being a non-model organism with a protracted and slow growth cycle, *Morus alba* lacks well-established genetic manipulation tools, rendering genetic screening a formidable challenge at this stage. On the other hand, even when gene knockout procedures are successfully executed, the presence of homologous genes poses a substantial obstacle to in vivo characterizing the function of MaDA. The elimination of

a single gene may not yield a significant impact on the production of Diels-Alder (D-A) products in *Morus alba*, as other genes may persist in performing analogous functions.

3. Regarding the non-stereoselective Diels-Alder reaction, the authors should clearly show the transition state energies for both transformations instead of Figure 5c, which makes it hard to understand what you show. They have to show the computational results such as Figure 4b in reference number 17 in this manuscript or Figure 4b in Sato, M., et al. Nature Catalysis, 2021, 4, 223-232.

Response #1-3:

In appreciation of the insightful recommendations from the reviewer, our earlier investigation revealed that in the absence of catalytic residues, the activation energies (ΔG^\ddagger) for *endo*- and *exo*-transition states (TS) were 22.2 and 23.7 kcal/mol, respectively. With the presence of the catalytic residue R443, these activation energies decreased to 20.3 kcal/mol for *endo*-TS and 20.8 kcal/mol for *exo*-TS (**Figure R3**) (Gao, L. et al. Nat. Catal. **2021**, 4, 1059). Thus, in the presence of the catalytic residues, both *endo*- and *exo*-pathways are kinetically favored, while the *endo*-pathway is theoretically more preferred than the *exo*-pathway since it features a slightly lower activation energy.

Figure R3. Theozyme calculations of MaDA–TS(*endo*) and MaDA–3–TS(*exo*) were reported in our previous work.

Different from the theozyme calculations, not all MaDAs can only produce the kinetically favored *endo*-product. Different MaDAs show opposite *endo/exo* selectivity such as MaDA and MaDA3. Our previous computational results suggested that *endo*-selective MaDAs (e.g., MaDA) could only stabilize the *endo*-TS rather than *exo*-TS while *exo*-selective MaDAs (e.g., MaDA3) could only stabilize the *exo*-TS rather than *endo*-TS (**Figure R4**, cited from Gao, L. et al. Nat. Catal. **2021**, 4, 1059), indicating that the different ability of MaDAs to stabilize the different transition states is another key regulator for the *endo/exo* selectivity.

Figure R4. Radial distribution functions (RDFs) from molecular dynamics trajectory. **a**, Histogram showing distributions of the distance (Å) between the reactive centre of MaDA-TS(*endo*) and FAD in the MD simulations. **b**, Histogram showing distributions of the distance between the reactive center of MaDA-3-TS(*exo*) and FAD in the MD simulation.

This study found that ancDA and ancDADS-mut4 contain the catalytic R443, which also engages in hydrogen bonding with the dienophile substrate's carbonyl group and stabilizes the *endo*-TS (Figures **3c** and **5d**). In contrast, ancDADS contains a His at this position, and thus could not stabilize the transition state and catalyze the corresponding D-A reaction. On the other hand, our MD simulations further demonstrated that after the residue substitutions involving H418R, D389E, A357L, and S348L, ancDADS obtained the ability to interact with the *endo* transition state just like MaDA1 and ancDA (**Figure 5c**). These results, together with our previous data in the Nature Catalysis paper, have elucidated the catalytic mechanism of ancDADS-mut4 in mediating the D-A reaction and providing additional support for the importance of this mutation.

Since the required calculations have already been reported in our previous papers (Gao, L. et al. Nat. Catal. **2021**, 4, 1059), we decided to cite the paper in the corresponding sentence instead of showing the data again.

4. The section on antifungal and antibacterial activity of Diels-Alder-type adducts should be deleted in this manuscript because it does not matter with Diels-Alderase. The authors should focus on demonstrating the intermolecular Diels-Alder reaction by MaDA.

Response #1-4:

We sincerely thank the reviewer for providing valuable feedback and conducting a meticulous review of the manuscript. We sincerely appreciate and acknowledge the

reviewer's insightful suggestions. We have deleted this section and revised it accordingly.

Response to Reviewer #2

Reviewer #2 (Remarks to the Author):

Comments: This study seeks to learn how Diels-Alderases (DAs) in Moraceae plants evolved their mechanisms. Evolutionary mechanisms showed these enzymes emerged from an ancestor with oxidocyclase activity. The authors use phylogenetics, enzymatic assays, structural studies, MD simulations and mutagenesis to study ancestral and modern variants to identify amino acids that drove the acquisition of Diels-Alderase activity and substrate binding. These studies are rigorous and provide mechanistic insight into how this family of enzymes evolved, while also giving insight into how proteins can be rationally engineered to achieve new catalytic functions. A few comments are provided below to help to improve the manuscript. After these comments are addressed, acceptance is recommended.

We thank this reviewer for dedicating time to assess our manuscript and providing positive feedback on our work. We have successfully addressed all the suggestions and comments raised by this reviewer.

1. This is a semantic matter: authors should consider replacing the word mutations/mutagenesis in connection to evolutionary changes with 'substitution'.

Response #2-1:

We sincerely appreciate the excellent suggestion from this reviewer and have revised it accordingly.

2. Authors state that the ionic hydrogen bond involving E389 and R418 promotes the D-A reaction by lowering the LUMO energy. It is not clear how this determination is made, as no energy calculation is reported in the manuscript. The authors should provide an explanation of this result or a reference explaining how this conclusion was arrived at.

Response #2-2:

Many thanks to this reviewer for highlighting this aspect. In our prior investigations (Gao, L. et al. Nat. Catal. **2021**, *4*, 1059), utilizing DFT theozyme calculations, we have discerned that R443 plays a pivotal role in activating the dienophile through an ionic hydrogen bond interaction, exhibiting an interaction energy of $-14.4 \text{ kcal mol}^{-1}$. This interaction significantly diminishes the gap between the highest occupied molecular

orbital (HOMO) of the diene and the lowest unoccupied molecular orbital (LUMO) of the dienophile by 0.8 eV (**Figure R5**).

The binding model analysis of MaDA–TS(*endo*) reveals a hydrogen bond triad involving Y412, E414, and R443, with R443 establishing contact with the carbonyl oxygen of the dienophile (**Figure R6**). Site-directed mutation studies suggest that, among these residues, Y412 holds relatively less significance than E414 and R443 (**Figure R6**). Consequently, E414 and R443 emerge as crucial contributors to facilitating the Diels-Alder reaction by reducing the LUMO energy in MaDA.

Figure R5. Orbital analysis. **a**, Orbital analysis for the LUMOs of the dienophile, **b**, Orbital analysis for the HOMO of the diene, **c**, Orbital analysis for the LUMO of the dienophile bound to R443 (MaDA), **d**, Orbital analysis for the dienophile bound to R294(MaDA-3). Our previous paper cited this data (Gao, L. et al. Nat. Catal. **2021**, *4*, 1059).

Figure R6. The binding model of MaDA–TS(endo). Data from our previous work (Gao, L. et al. Nat. Catal. **2021**, 4, 1059).

In our study, residues E414 and R443 in MaDA correspond to E389 and R418 in ancDA, respectively. Sequence alignment reveals that both E389 and R418 are highly conserved across all DAases, including the ancestral DAase ancDA (**Figure 3c and 3d**). Therefore, the conclusion drawn in this research is equally applicable to homologous genes of the DAases investigated and the ancestral DAase. Specifically, E389 and R418 facilitate the occurrence of the Diels-Alder reaction by lowering the energy of the dienophile's LUMO.

To convey the aforementioned information more clearly, we have rephrased this section in the manuscript as follows:

“Drawing from our previous research, we observed that in MaDA, R443 contributes to lowering the dienophile's LUMO energy by 0.8 eV through hydrogen bonding interactions, facilitating the Diels-Alder reaction. Additionally, E414 stabilizes these interactions by forming a hydrogen bond with R443³⁰. As similar interactions were also noted in ancDA, we hypothesise that R418 and E389 in ancDA play analogous roles in catalysing the Diels-Alder reaction, similar to the functions of R443 and E414 in MaDA (Fig. 3c).”

3. Authors have used MD simulations to determine that substrates or transitions do or do not bind to enzymes. However, the only data they show to support this claim is distances, which is not enough. Distances can be used to support a more sustained interaction between components of the protein-TS complex, but they cannot be used

to describe tight binding (or even binding), unless binding affinities have been calculated, which can be done from MD simulations. Authors should either perform binding calculations or remove the quantitative language about binding, replacing it with something like ‘transition state maintains an interaction with the active site over the course of our simulation’. This comment refers to data shown in Figures 5C and 5D

Response #2-3:

We greatly appreciate the insightful comments provided by the reviewer. As the reviewer suggested, we have removed the quantitative language about binding and rephrased the related sentences as follows:

Change “our MD simulations showed that the transition state for the intermolecular D-A reaction between diene 3 and morachalcone A (4) could not bind to the active site of ancDADS (Fig. 5c)” to “our MD simulations showed that the transition state for the intermolecular D-A reaction between diene 3 and morachalcone A (4) could not maintain a stable interaction with the active site of ancDADS during the simulations (Fig. 5c).”

Change “We found that the transition state could also tightly bind to the active site of ancDADS-mut4 like ancDA and MaDA1 (Fig. 5c).” to “In contrast, the transition state maintains an interaction with the active site of ancDADS-mut4 as well as ancDA and MaDA1 over the course of our simulation (Fig. 5c).”

4. Because models were prepared by AlphaFold, authors should consider providing the confidence values (per-residue) of their structures. This will provide a sense of how reliable the models are.

Response #2-4:

We appreciate the suggestions provided by the reviewer. Following your advice, we have generated structural diagrams of proteins predicted by AlphaFold, employing colorization based on the numerical values of per-residue confidence score (pLDDT) (**Figure R7**). The results reveal that the pLDDT values for over 90% of the residues

exceed 80, with values exceeding 90% for over 80% of the residues. Most pLDDT values are notably high, with only a few regions exhibiting lower scores. Furthermore, the predicted protein structures show a fundamental consistency with the backbone of structures resolved through crystallography. These findings collectively indicate the fundamental reliability of AlphaFold in predicting protein structures. The detailed results are shown in the supplementary data in **Figure 1** of the manuscript.

Figure R7. Structure prediction with AlphaFold. **a-d**, Predicted structures for ancDA, ancDADS, ancDADS-mut4, and MaDA are presented. Residues are color-coded based on their per-residue confidence score (pLDDT), ranging from the minimum value to the maximum value. Lower pLDDT values indicate lower confidence, while higher values signify more confident predictions. **e**, An alignment is shown between the

predicted structure (light purple) of MaDA and the crystal structure of MaDA (cyan, PDB ID 6JQH).

5. Critical details of the MD simulation are missing. Was solvent added, and how much? Were ions added? Were minimization and equilibration performed? What type of ensemble, thermostat etc were used.

Response #2-5:

We thank the reviewer for raising important queries regarding the critical details of our Molecular Dynamics (MD) simulations. In response, we have provided the required details in the method section as follows:

“For the molecular dynamics (MD) simulations, each docking complex featuring the transition state (TS) with ancDA, ancDADS, and ancDADS-mut4 served as the foundational structure. Protonation states were determined using the Protein Preparation Wizard in Maestro. We employed the OPLS4 force field parameter set for both the protein and ligand, accompanied by the TIP3P model for water.

Each protein complex was immersed in a pre-equilibrated orthorhombic box with a 10 Å buffer of TIP3P water molecules, and the volume was minimized. Explicit counterions (Na⁺ or Cl⁻) were added to neutralize the systems, and 0.15 M NaCl was introduced to simulate ionic effects in a realistic environment.

Molecular dynamics simulations were conducted using the Desmond software package, employing a standard NPT (isothermal-isobaric ensemble) relaxation protocol. The simulation protocol unfolded in multiple stages to explore the system's behavior under diverse conditions. All time units are reported in picoseconds (ps); energy is expressed in kilocalories per mole (kcal/mol).

Utilizing default parameters, the simulation was performed with a single CPU, employing a cutoff radius of 9.0 Å, a time step of [0.002 0.002 0.006] ps, and a total simulation time of 500,000 ps. For temperature control, the Berendsen algorithm was employed with a target temperature of 343.0 K and a coupling time of 1.0 ps. Pressure control was achieved using the NPT ensemble with a pressure setting of 1.01325 bar. The pressure coupling time was 2.0 ps, and the temperature coupling time was 1.0 ps.

The restraint protocol employed a force constant of 50 kcal/mol/Å² on heavy atoms. To define the transition state geometry, a harmonic constraint with a force constant of 500 kcal mol⁻¹ Å⁻² was applied. Torsional constraints were applied to the planes of diene and dienophile carbons with a force constant of 500 kcal mol⁻¹ Å⁻². These restraint parameters were selected to ensure the preservation of the desired structural features during the simulation while allowing for the exploration of dynamic behavior.”

We trust that these clarifications provide a more detailed understanding of our MD simulation methodology.

Response to Reviewer #3

Reviewer #3 (Remarks to the Author):

Comments: Lei et al. report an interesting study to trace the evolutionary origin of the intermolecular Diels-Alderase and their functionally coupled dehydrogenases in *Morus alba* (named MaDAs and MaDSs, respectively). Based on a thorough phylogenetic analysis with an association of enzymatic activity assays, the authors raised the hypotheses that both MaDAs and MaDSs share with the oxidative cyclases in Moraceae plants (named MaOCs) an ancestor enzyme possessing a berberine bridge enzyme (BBE)-like fold and that OC activity appears to be more ancestral and can be developed to individual dedicated OCs, Das and DSs through gene duplication and mutagenesis. To validate these hypotheses, they constructed the ancestral Diels-Alderase (ancDA) and the last common ancestor of DAs and DSs (ancDADS) for comparative analysis. Consequently, the ancDADS, which basically is an OC and has no DA and DS activities, was developed to an enzyme with DA activity via a minimum of the mutations of four key residues. Eventually, computational studies rationalized the catalytic processes for ancDA, ancDADS and ancDADS-derived enzymes with DA activity, leading to a model for the evolution of OC to DA.

Overall, this is a very interesting and comprehensive study that showcases how enzymes evolve for functional differentiation to process structurally related substrates in different ways by combining genetic, biochemical, structural biology and computational approaches together. The manuscript is well-written. The experiments are clearly and comprehensibly presented, and figures are well-arranged. Understanding the evolutionary origin and mechanism of MaDAs will ultimately facilitate design and develop new enzyme tools for synthetic biology utilization. This reviewer is supportive of publication (although minor revision appears to be necessary) because the results should be of great interest to the interdisciplinary readership of the journal.

We appreciate the very supportive comments from this reviewer.

1. Enzymatic activity assays of selected MaOCs, MaDAs and MaDSs revealed that a few are functionally promiscuous, i.e., two DAs and two DSs have OC activity for moracin D formation. Did the authors determine whether DAs have DS activity or DSs have DA activity? If not, could you provide an explanation, particularly based on the evolutionary model shown Fig. 2?

Response #3-1:

We greatly appreciate the insightful query raised by the reviewer. When the diene **3** and morachalcone A (**4**) were added into the buffer containing diene synthases, no detectable D-A product was observed (**Figure R8b**), indicating that the diene synthase in mulberry tree cannot catalyze the D-A reaction. Meanwhile, we also tested whether MaDAs could catalyze the diene formation from moracin C and found all the characterized MaDAs could not catalyze diene formation (**Figure R8c**). These results indicated that MaDAs and MaDSs independently evolved from MaOCs, further supporting our evolutionary model shown in Figure 2d.

Figure R8. The LC-MS results for Diels-Alder reactions and diene-synthase reactions. **a**, The figure showcases the catalytic reactions facilitated by DAs, DSs, and OCs. **b**, Ultra-performance liquid chromatography (UPLC) analysis of the Diels-Alder reactions catalyzed by DSs. MaDA was used as a positive control. **c**, UPLC analysis of the diene formation reactions catalyzed by DAs. The diene synthase MaMO was used as a positive control.

2. Page 7. The single mutation of I417L in ancDA resulted in the complete loss of DA activity. This result appears to be a bit contradictory to the weak DA activity of ancDADS-mut4 in which the I417L mutation is not involved.

Response #3-2:

Thank this reviewer for his/her thorough review of our manuscript and for providing valuable feedback. We acknowledge the concern raised regarding potential inconsistencies in our findings and would like to offer clarification to ensure a nuanced interpretation of our research.

The relative enzymatic activities shown in Figure 3e were determined at 70 degrees for 7 minutes, and no D-A products were observed for ancDA-L348S, ancDA-L357A, ancDA-E389D, ancDA-L417I, ancDA-R418H variants, indicating the critical role of these residues for the DA activity. However, upon extending the reaction time from 7 minutes to 2 hours, ancDA-L417I exhibited D-A activity while the other four variants did not (**Figure R9**). These results suggest that compared with the other 4 substitutions, L417I substitution has less influence on the D-A activity of ancDA. Since I417L substitution was found to be important but not essential to maintain the DA activity of ancDA, it is consistent with our mutagenesis results that ancDADS obtained weak DA activity after the introduction of L348S, L357A, E389D, and R418H substitutions and the activity of ancDADS-mut4 was further improved by the introduction of L417I substitution (Figure 4).

Figure R9. D-A activity of the ancDA mutants after prolonged reaction time.

In the revised version of our manuscript, we have discussed these updated results in the manuscript and added the data (**Figure R9**) into the **Supplementary Data Figure 2**. The newly added description contains “Moreover, further enzymatic assays with longer incubation time also suggested that L417I has less negative influence on the DA activity of ancDA compared with the other 4 substitutions (e.g., L348S, L357A, E389D, and R418H) (**Supplementary Data Fig. 2**). Therefore, L417 is an important but not necessary substitution in the emergence of DA activity” and has been put in the last paragraph page 7.

3. Extended Data Fig. 7, the lane for the incubation of ancDADS-mut5 with substrates 3 and 4. The product profile is complicated and have some peaks in addition to those indicated. What do these peaks represent? It seems additional activities to OC and DA activities were observed?

Response #3-3:

Thank this reviewer for his/her thorough review of our study and the valuable suggestions he/she provided. Regarding his/her inquiry about the extensive reaction products observed in the incubation of ancDADS-mut5 with substrates 3 and 4 in **Extended Data Fig. 7**, we have conducted additional analyses to identify the origin of these additional peaks. We are pleased to share our observations and explanations with this reviewer.

Figure R10. The initial UPLC-MS analysis results of ancDADS mutants.

The initial graph is shown in **Figure R10**, with major unidentified peaks highlighted with asterisks. Since diene 3 was an unstable compound, we typically generated diene 3 *in situ* by hydrolysis of its tri-acetylated precursor and directly used the reaction mixture

in the enzymatic assays. After carefully checking the UV spectra and MS signals of those peaks, we think these additional peaks correspond to incomplete deprotection products during the *in situ* hydrolysis of the diene **3** (Figure R11).

Figure R11. MS spectra and UV absorption peaks of miscellaneous peaks. a, The UV and MS spectra of the peaks have a retention time of around 5.60 and 5.70 minutes, respectively. **b,** The UV and MS spectra of the peaks have a retention time of around 5.10 and 5.20 minutes. **c,** The UV and MS spectra of the peak corresponding to the diene **3**.

Based on these results, we repeated the hydrolysis reaction to ensure the complete hydrolysis of the diene precursor. We performed the enzymatic assays of ancDADS variants using the relative pure diene **3**. After repeating these experiments, these peaks disappeared (Figure R12) and we have updated the revised UPLC spectra data in the **Extended Data Fig. 7**.

Figure R12. The revised UPLC-MS analysis of ancDADS variants.

4. Chemically, the evolution of an OC from a DS is possible, given the notion that cyclization activity can be developed for 1,4-addition by following dehydrogenation immediately. Thus, the mechanism by which MaOCs function is interesting: during the catalysis, does cyclization occur prior to dehydrogenation?

Response #3-4:

As the reviewer pointed out, two possible mechanisms exist for the MaOCs-catalysed oxidative cyclization (**Figure R13a and R13b**). The first involves forming an unstable quinone intermediate A which may spontaneously cyclize to form the 2-H pyran ring via 1,4-addition or 6 π cyclization reaction. On the other hand, the secondary mechanism involves the formation of a relatively stable intermediate B, which MaOCs will further oxidize to form the double bond. Although further studies, such as DFT calculations, are desperately needed to fully elucidate the mechanism, some evidence

from the literature and our experiments suggests that the first one is probably the catalytic mechanism of MaOCs. They are as follows:

- 1) There have been some studies on the catalytic mechanism of BBE-like enzymes, among which THCAS (Tetrahydrocannabinolic acid synthase) catalyzes a very similar reaction as MaOCs (**Figure R13c**). The reaction is initiated by a concerted hydride transfer to FAD and deprotonation of the hydroxyl group of substrate CBGA by Tyr484 to give the quinone intermediate in a suitable conformation. Subsequently, the intermediate would undergo a cyclization leading to THCA (J. Mol. Biol. **2012**, 423, 96-105).
- 2) we have never detected any intermediate in the MaOCs-catalysed oxidation of moracin C. If the stable intermediate B is generated in the enzymatic reaction, we can probably observe this intermediate in UPLC-MS analysis.

Figure R13, Proposed mechanisms of MaOCs and their homologous protein THCAS. a, The first proposed mechanism of MaOCs. b, The second proposed

mechanism of MaOCs. c, Proposed mechanism of THCAS based on crystal structure and mutational analysis.

Minor issues:

5. Page 2, abstract, line 35: “plant-derived Diels-Alder-type secondary metabolites” . It could be better to specialize to “moraceae plant-derived” .

Response #3-5:

We have refined the phrase " plant-derived Diels-Alder-type secondary metabolites" to "Moraceae plant-derived Diels-Alder-type secondary metabolites "

6. Page 2, abstract, line 40: please remove the comma in “D389E, and H418R” and in “and H418R, that alter” .

Response #3-6:

We appreciate the feedback. We have adjusted the expression " D389E, and H418R " to " D389E and H418R ".

7. Page 3, line 83: please remove the first “typically” in “Unlike most BBE-like enzymes typically, which typically catalyze” .

Response #3-7:

Thank you for your correction. We have eliminated the redundant "typically" and revised the phrase to "Unlike most BBE-like enzymes, which typically catalyze."

8. Page 4, line 100: “FAD-dependent DAs in Moraceae plants evolve from the BBE-like enzymes” could be better.

Response #3-8:

Acknowledged. We have revised the sentence to “FAD-dependent DAs in Moraceae plants evolve from the BBE-like enzymes”.

9. Page 6, line 177: please leave a blank place between “MaDA1(Extended Data Fig. 5a” .

Response #3-9:

Noted. We have inserted a space and modified the text to "MaDA1 (Extended Data Fig. 5a)."

10. Page 7, line 207: please leave a blank place between "mongolicin F(exo-7)" .

Response #3-10:

Understood. We have added a space, and it now reads "mongolicin F (exo-7)."

11. Page 10, Line 318: the sentence “ ” needs to be rephrased. It could be better “ This evolutionary strategy represents a rare case for metabolic pathway development, by evolving a same ancestor to two new enzymes catalysing two consecutive reactions” .

Response #3-11:

Thank you for your valuable input. We have rephrased the sentence to “This evolutionary strategy represents a rare case for metabolic pathway development, by evolving a same ancestor to two new enzymes catalysing two consecutive reactions”.

Response to Reviewer #4

Reviewer #4 (Remarks to the Author):

Comments: Although many enzyme-catalyzed Diels-Alder reactions have been identified, the evolutionary mechanisms of natural Diels-Alder enzymes remain unclear. Lei and co-authors use phylogenetic analysis, ancestral sequence reconstruction, and other methods to elucidate the evolutionary origin of Diels-Alder enzymes in Moraceae plants. They found that these Diels-Alder enzymes originated from gene duplication followed by neofunctionalization of BBE-like enzymes with oxidocyclase activity. The results are interesting and novel to the natural product biosynthesis research.

We acknowledge the valuable feedback provided by the reviewer. The manuscript has been meticulously revised, incorporating all the suggested improvements and comments.

1. This work involved docking and Molecular dynamics simulation analyses to investigate the substrate binding pattern change in the evolutionary process from OCs to DAs. It might be better to perform quasi-classical direct-dynamics simulations as Ken Houk and co-authors did (J Am Chem Soc, 2016, 138, 3631-3634).

Response #4-1:

We greatly appreciate this reviewer for his/her concerns. We appreciate this reviewer's recommendation regarding quasi-classical direct-dynamics simulations; however, we believe that traditional molecular dynamics simulations are more suitable for our study for the following reasons:

Firstly, our research focuses on the change in substrate binding patterns during the evolutionary process from OCs to DAs, which involves a relatively longer time scale. Traditional molecular dynamics simulations excel in handling dynamic changes over larger time scales, particularly in understanding the evolutionary pathways and stable conformations of complex systems.

Secondly, quasi-classical direct-dynamics simulations are generally used to unravel the inherited complex dynamic of chemical reactions involving multiple transition states or intermediates through classical simulation of the trajectories (Patel, A. et al. J. Am. Chem. Soc., **2016**, 138, 3631-3634). Due to the complexity of enzymes, this type of calculation for enzyme-catalysed reactions is typically performed using a simplified enzyme-free model (Soler, J. et al. J. Am. Chem. Soc. **2022**, 144, 15954–15968), which can not reveal the dynamic interactions between enzymes and the transition

states. Since the inherited dynamics of the enzyme-independent Diels-Alder reactions have been well established in our previous paper (Gao, L et al. *Nat. Chem.*, **2020**, *12*, 620; *Nat. Catal.* **2021**, *4*, 1059), this work is aimed to elucidate how the mutations in MaOCs contributes to the emergence of DA activity and thus molecular dynamics simulations were used to achieve this goal.

2. In line 70 of page 3, the authors wrote that “the monofunctional DAs PyrE3, SdnG and SpnF”. However, the work of Hung-wen Liu and Ken Houk showed that SpnF is a multifunctional enzyme with [4+2], [6+4], and cope rearrangement activities. (*J Am Chem Soc*, 2016, *138*, 3631-3634; *PNAS*, 2017, *114*, 10408-10413.)

Response #4-2:

Thank you for your insightful suggestion. We have revised this paragraph accordingly and cited the above-mentioned two references in the sentence.

3. In line 360 of page 11, the authors wrote, “Protein sequences with fewer than 450 or more than 650 amino acids were removed.” Would this prevent the identification of new Diels-Alder enzymes?

Response #4-3:

We thank the reviewer for their insightful question and meticulous review of our manuscript. In response to this reviewer’s inquiry about the exclusion criteria mentioned on page 11, line 360, we would like to clarify why the protein sequences with fewer than 450 or more than 650 amino acids were ignored in our experiments.

The FAD-dependent Diels-Alderase (DAs) in *Morus* plants are members of the BBE-like enzyme protein family, distinguished by two distinct domains, namely FAD_binding_4 (PF01565.23) and BBE (PF08031.12) (Daniel, B. et al. *Arch. Biochem. Biophys.* **2017**, *632*, 88). Figure R14 provides detailed domain information for the representative Diels-Alderase MaDA. Therefore, a functionally normal Diels-Alderase typically contains around 550 amino acids and these two conserved domains.

Figure R14. The domain structure of MaDA. The numbers indicate the position of the amino acid in the protein.

There are 27,473 protein sequences in the genome of *Morus alba*; considering that some protein sequences might be incomplete due to insufficient coverage and depth of genome sequencing, we merged the sets of proteins containing either the FAD_binding_4 domain or BBE domain to avoid the omission of novel Diels-alderases, resulting in 64 proteins with lengths ranging from 90 to 590 amino acids (**Figure R15a**). We then constructed an initial phylogenetic tree based on these 64 proteins as well as several other functionally characterized BBE-like enzymes as the outgroup (**Figure R16**). Subsequently, all the candidate Diels-Alderase in *Morus alba* were identified which are in the same subgroup with the previously characterized Diels-Alderase MaDA (highlighted in red and green in **Figure R16**). The protein sequence lengths of candidate Diels-Alderase range from 168 to 570, as illustrated in the frequency histograms below (**Figure R15b**).

Figure R15. Frequency histograms of the lengths of the identified BBE-like enzymes. **a**, Frequency histogram of all proteins before constructing the initial phylogenetic tree. **b**, Frequency histogram of proteins after filtering following phylogenetic tree construction.

To ensure a comprehensive function characterization of the candidate DAs, we strategically selected one candidate gene from each branch of the initial phylogenetic tree, considering sequence consistency and functional proximity (**Figure R16**). Preference was given to sequences with complete lengths on the same branch to facilitate primer design for experimental validation. Therefore, to identify novel Diels-Alderase and related BBE-like enzymes in *Morus alba*, the protein sequences with fewer than 450 or more than 650 amino acids were not initially removed in the sequence mining process. They were not chosen for the following gene cloning and function characterization. After eliminating identical genes (>95% identity) amplified from the genome or cDNA of *Morus alba*, the phylogenetic tree of the identified BBE-like enzymes in *Morus alba* was constructed and depicted in Figure 2b.

Since all the newly characterized BBE-like enzymes including MaDAs from *Morus alba* have a protein sequence length ranging from 450 to 650 amino acids, this empirical feature, rooted in our experience, guided the genome mining of novel Diels-Alderases in other species and the construction of the updated phylogenetic tree for the DAs and oxidases (Extended Data Fig. 2). The chosen length criteria resulted in a phylogenetic tree with high Bootstrap values, minimizing inaccuracies that may arise from the extended length span of the initial phylogenetic tree. In this case, as the reviewer pointed out, novel Diels-Alderases from other species might be missed if their complete protein sequences are unavailable in the database.

To present this filtering process more comprehensively and clearly, we have deleted the sentence “protein sequences with fewer than 450 or more than 650 amino acids were removed” in the sequence mining section and added the description in the phylogenetic analysis section as follows:

' For candidate genes from other species, the protein sequences with fewer than 450 or more than 650 amino acids were removed to enhance the accuracy of the phylogenetic tree.'

Figure R16. The initial phylogenetic tree of the identified BBE-like enzymes. The genes selected for experimental validation are highlighted in bold and marked with blue stars.

4. In line 430 of page, "0. 0.97915 Å" should be replaced by "0.97915 Å".

Response #4-4:

Thank you for your correction. We have revised the sentence "0. 0.97915 Å" to "0.97915 Å."

REVIEWERS' COMMENTS

Reviewer #1 (Remarks to the Author):

This reviewer was convinced the changes in the revised manuscript.

The paper is suitable and recommended for publication in Nature Communications as an Article in the current form.

Reviewer #2 (Remarks to the Author):

Authors have addressed my comments satisfactorily. Publication is recommended.

Reviewer #3 (Remarks to the Author):

The authors have satisfactorily addressed the issues from the reviewers. The manuscript is now believed to be ready for publication.

[Editorial note]

Reviewer 3 has assessed authors' responses to Reviewer 4 and considers them addressed.